# Growing with Your Embodied Agent: A Human-in-the-Loop Lifelong Code Generation Framework for Long-Horizon Manipulation Skills

## Abstract

Large language models (LLMs)-based code generation for robotic manipulation has recently shown promise by directly translating human instructions into executable code, but existing approaches are limited by language ambiguity, noisy outputs, and limited context windows, which makes long-horizon tasks hard to solve. While closed-loop feedback has been explored, approaches that rely solely on LLM guidance frequently fail in extremely long-horizon scenarios due to LLMs' limited reasoning capability in the robotic domain, where such issues are often simple for humans to identify. Moreover, corrected knowledge is often stored in improper formats, restricting generalization and causing catastrophic forgetting, which highlights the need for learning reusable and extendable skills. To address these issues, we propose a human-in-the-loop lifelong skill learning and code generation framework that encodes feedback into reusable skills and extends their functionality over time. An external memory with Retrieval-Augmented Generation and a hint mechanism supports dynamic reuse, enabling robust performance on long-horizon tasks. Experiments on Ravens, Franka Kitchen, and MetaWorld, as well as real-world settings, show that our framework achieves a 0.93 success rate (up to 27% higher than baselines) and a 42% efficiency improvement in feedback rounds. It can robustly solve extremely long-horizon tasks such as "build a house", which requires planning over 20 primitives.[1]

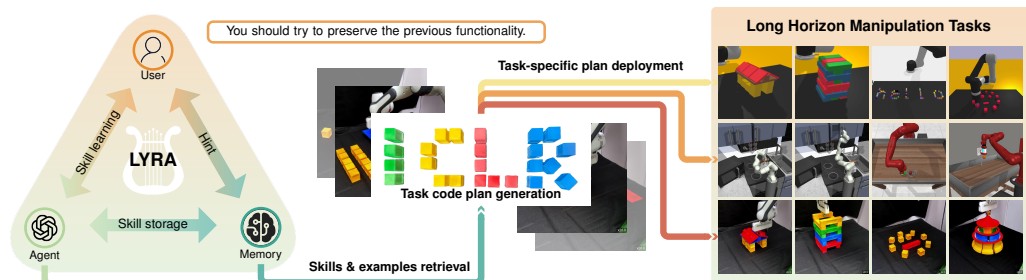

Figure 1: Framework overview: human-in-the-loop lifelong skill learning and task deployment.

## 1 Introduction

Large language models (LLMs) and vision-language models (VLMs) have become integral to robotic manipulation due to their robust commonsense knowledge and advanced reasoning capabilities. Early approaches Co-Reyes et al. (2018); Lynch et al. (2023); Liu et al. (2023); Meng et al. (2025a) relied on language embeddings conditioned within reinforcement learning or imitation learning to align robot actions with human commands. These methods often struggled with limited

---

[1]Code will be open-sourced upon acceptance.

data efficiency and poor generalization. With the rapid progress of LLMs such as GPT, a natural direction has been to integrate them into the pipeline for task decomposition and language grounding Zhang et al. (2023b); Huang et al. (2023); Guo et al. (2024). In this setting, an LLM decomposes a complex manipulation task into sub-tasks and invokes a pre-trained language-conditioned policy to execute low-level primitives. This approach assumes that the pre-trained policy can carry out each motion precisely, yet in practice, this is rarely possible due to environmental perturbations and imperfect policy design. Another direction for advancing human-level robotic manipulation is to adopt LLM or VLM backbones for large-scale pretraining on robotic data, creating end-to-end vision–language–action (VLA) foundation models Kim et al. (2024); Black et al. (2024); Bjorck et al. (2025). However, robotic data is far more limited than in computer vision or natural language, leading to data sparsity. Training such large models (often with at least 7B parameters) also demands enormous computational resources and makes deployment on edge devices difficult. These factors have slowed progress in this line of research. Alternatively, well-pretrained LLMs already possess strong instruction-following capabilities and can directly generate policy code as an action representation for robot control.

Using LLMs for code generation in robotic manipulation Zhang et al. (2025) has shown strong potential in embodied AI. Approaches like Code-as-Policies (CaP) Liang et al. (2023); Chen et al. (2024); Mu et al. (2024) translate human instructions into executable Python code with fixed sets of perception and control primitives, enabling rapid deployment of behaviors. Despite this progress, current methods face key limitations: (1) LLM outputs are often noisy and error-prone in long-horizon planning. (2) Natural language is inherently ambiguous, so the agent may not always capture the user's intention. (3) Agents are restricted by pre-defined primitives and handcrafted prompts, and the limited context window prevents scaling with many examples. As a result, agents struggle with complex long-horizon tasks. Prior works Liang et al. (2024); Zhi et al. (2025); Meng et al. (2025b) incorporate feedback to improve robustness, but these do not constitute true "learning", since verifiers must repeat the same feedback for seen tasks in the future. In code generation, learning can be enabled by dynamically adjusting prompts to support LLM in-context adaptation. This raises two key questions. The first question is **how to store and reuse knowledge from feedback.** Updating the prompt directly Arenas et al. (2024) can cause catastrophic forgetting, where performance on earlier tasks drops sharply. Extracting insights across tasks Zhao et al. (2024); Zha et al. (2024) produces ambiguous, task-specific knowledge, lacking in generalization to unseen tasks. Saving flat code offers flexibility for modification, but repeated changes often introduce noise and errors, making long-horizon generation unstable. A more stable alternative is to encapsulate temporally extended behaviors as **skill functions**, where adjusting function parameters offers both flexibility and generalization while preserving stability. This makes skills a promising choice for storing feedback knowledge. The second question is **what type of feedback is most efficient during interaction.** Language feedback is a convenient interface, but feedback generated solely by LLMs Arenas et al. (2024); Wang et al. (2023); Meng et al. (2025b) is often unreliable, as it may diverge from human preferences or fail in extremely long-horizon reasoning Wang et al. (2023). Human-provided feedback, in contrast, is more robust: errors in robotic tasks are usually easy for humans to identify, evaluate, and correct. Moreover, the feedback can be tested and verified within seconds in robotic code generation, whereas training traditional deep learning methods often requires hours or even days. This reliability and efficiency highlight the unique advantage of **involving humans to guide robotic skill learning**.

Motivated by these questions, we propose a human-in-the-loop framework that encodes user feedback into reusable skills and extends their functionality through a user-designed curriculum, enabling preference-aligned lifelong skill learning (Fig. 1). Learned skills and examples are stored in an external memory to prevent catastrophic forgetting and support reuse. For long-horizon task planning, our framework employs Retrieval-Augmented Generation (RAG) to retrieve relevant skills and examples, enabling dynamic in-context learning. In addition, a simple yet effective **hint** mechanism allows users to guide the agent when retrieval alone is insufficient, ensuring that only the most relevant skills are applied, thereby improving efficiency and reducing interference from irrelevant data. Our central contribution is demonstrating that human feedback can be distilled into stable, generalizable, and compositional skill representations that persist throughout lifelong use, enabling robust long-horizon manipulation beyond the capabilities of fully automated LLM-based systems. We validate the framework through experiments in Ravens, Franka-Kitchen, and MetaWorld, as well as real-world evaluations on a Franka FR3 across diverse long-horizon tasks. Empirical analysis shows a 0.93 success rate, up to 27% higher than baselines, and a 42% efficiency improvement

in correction rounds compared with LLM-based closed-loop methods. Notably, the framework can robustly solve extremely long-horizon tasks such as "build a house", which requires planning over 20 primitives.

To summarize, our contributions are threefold: (1) **Human-in-the-loop lifelong skill learning**: A framework that distills human feedback into stable, reusable, and compositional skills, and extends them through lifelong learning while preserving previously acquired functionalities. (2) **Memory-augmented skill retrieval**: An external memory that stores learned skills and examples, combined with RAG and user-provided "hints" for dynamic in-context learning. (3) **Challenging extreme long-horizon tasks**: Demonstrating the first successful solution to the "build a house" task requiring over 20 primitives, validated across both simulation and real-world settings. We call our framework **LYRA**: A **L**ifelong learning code s**Y**nthesis framework with human-in-the-loop for **R**obotic long-horizon skill **A**cquisition.

## 2 METHOD

In this section, we first introduce the preliminaries, then describe how our framework learns skills and applies them to downstream tasks.

### 2.1 PRELIMINARIES

Code generation for robotic manipulation can be modelled as using an LLM $f$ to map a natural language instruction $l$ to a robot control code $c$, i.e., $f(l) = c$, where $f$ is termed a Language Model Program (LMP) in prior work Liang et al. (2023). Each code $c$ is achieved by multiple behaviours $b$; thus, $f$ can also be viewed as a mapping from $l$ to $b$. A behaviour $b$ is defined as a desirable or semantically meaningful motion (e.g., picking up an object), and the set of all such behaviours is denoted by $\mathcal{B}$. In our work, we argue that behaviours $b$ requiring no variation can be encapsulated as *skills* that reliably elicit these behaviours. We denote the skill space as $\mathcal{Z}$ and state parameters as $s$. Importantly, a skill $z \in \mathcal{Z}$ may also build on other skills:

**Definition 1**: A *skill* $z \in \mathcal{Z}$ is a function $b \sim z(s)$ which induces a specific behaviour $b$.

It is worth noting that although the task code $c$ generated by the agent represents a sequence of behaviors to accomplish a task, it is not equivalent to a skill. We define a *task* $(l, s_0)$ by a natural language instruction $l$ and an initial environment state $s_0$, which together form the main interface between the user and the agent. When a task is successfully completed with user feedback, we extend this tuple to $(l, s_0, c, s_T)$, where $c$ is the task-specific code (often flat, process-oriented) and $s_T$ is the achieved final state. In practice, $c$ may call skills $z$ to complete the task, and its execution should produce an evaluable task result.

To align the skill output with the user's desired behaviour $b$, the agent adjusts the function details under user guidance. We call this process *learning the skill*. Importantly, here, learning does not refer to neural network fitting on data, but instead follows a broader concept in computer science Mitchell (2006):

**Definition 2**: An agent is said to *learn* if it produces better behavioural responses $b$ to an instruction $l$, as evaluated by a human.

Based on these definitions, the learning pipeline should answer: (1) How to encode human-provided feedback into skills that align with user preferences? (2) How to extend skills through lifelong learning while enabling dynamic in-context adaptation? (3) How to reuse and transfer learned skills to unseen long-horizon tasks?

### 2.2 PREFERENCE-ALIGNED SKILL ACQUISITION

Based on the above questions, we design a three-phase pipeline to preserve and utilize human-in-the-loop corrections for preference-aligned lifelong skill learning in long-horizon manipulation (Fig. 2). The phases are: (1) preference-aligned skill acquisition, (2) lifelong capability extension, and (3) task-specific retrieval and planning.

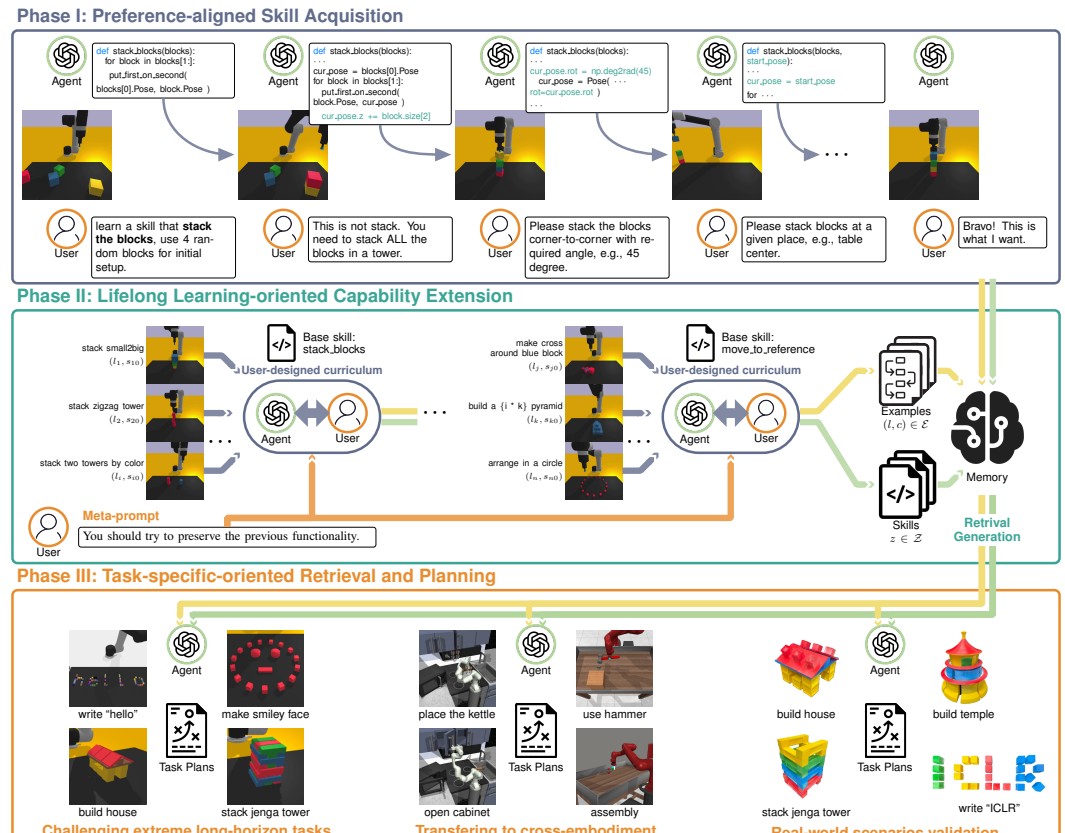

Figure 2: Structure of the proposed human-in-the-loop lifelong skill learning pipeline.

To learn a new skill, three elements must be clarified: (1) what skill to learn, (2) the initial task environment, and (3) the expected functionality or behavior. Asking an LLM to infer all of this at once is impractical due to language ambiguity, so we adopt multi-turn interaction. The agent is initiated with a fixed set of skills $z \in \mathcal{Z}_0$, along with examples $e = (l, c) \in \mathcal{E}_0$ that demonstrate their use, enabling basic environment interaction. The user first describes the skill in natural language, and the LLM invokes the "skill_parse" API to generate a Python function header with suggested parameters (e.g., "def stack_blocks(blocks, start_pose):"). The user can accept or refine this definition. Once accepted, the LLM requests a base environment setup, and generates a corresponding task scenario. The complexity of the task is user-controlled, but overly complex tasks (over 6–8 steps) hinder skill acquisition when the agent's ability is still limited. We therefore recommend two principles: (1) tasks should be simple, ideally completed within 1–4 primitives, and (2) tasks should allow flexibility so that one skill can adapt to diverse requirements. After the setup, the user and the LLM agent will work together to "learn" the skill. The agent generates candidate code, deploys it in simulation, and displays results in real time. The user then accepts, rejects, or provides free-form feedback to refine the skill. User tests multiple requirements within the same scenario to check if the skill and its parameters generalize. For example, in Fig. 2 Phase I, the user guides the agent to stack blocks under different conditions such as edge alignment, tower positioning, or rotation. During this phase, **the agent can freely modify implementation details under the reserved skill name**, with the objective of producing a functional, preference-aligned skill. This forms the foundation for later phases of lifelong skill learning and capability extension.

## 2.3 LIFELONG LEARNING-ORIENTED CAPABILITY EXTENSION

Learning skills from a single task scenario is insufficient for complex long-horizon tasks, for two main reasons: (1) Simple skills cannot handle variations beyond their learning distribution. For example, a skill like "stack_blocks" for stacking four blocks at a fixed pose cannot generalize to tasks such as building a $\{i \times j \times k\}$ structure or stacking multiple towers by color or size. (2) Directly

inserting learned skills into prompts risks hallucination, where LLMs may rewrite or fabricate code during long-horizon interactive planning, leading to catastrophic forgetting of existing skills.

Inspired by lifelong learning, we extend the pipeline with a user-designed curriculum that enables **bottom-up skill functionality expansion** (Fig. 2 Phase II). In this phase, the agent must: (1) solve more complex user-defined task variations, (2) preserve and extend skill functionality to unseen cases, and (3) store skills permanently for reuse. The learning process mirrors Phase I, with an added **meta-prompt that explicitly asks the agent to preserve prior functionality while adapting to new tasks**. Users should design tasks outside the original distribution and aligned with long-term goals, guiding the agent to acquire foundational skills for future complex tasks. The agent can either (1) create a new named skill that calls existing ones as nested functions, or (2) extend the current skill using modularized code (e.g., if-else or match-case) to avoid overwriting. After adapting to new tasks, the agent is re-evaluated on prior tasks; if performance is preserved, we can say that lifelong learning is achieved. To further prevent forgetting and hallucination, both learned skills $z$ and explored successful examples (including task instruction and task plan) $(l, c) \in \mathcal{E}$ are stored in **external memory** and retrieved via RAG during task planning.

## 2.4 Task-specific-oriented Retrieval and Planning

In Fig. 2 Phase III, we enable **long-horizon planning** through in-context learning, adapting general-purpose LLMs to specific tasks using two main prompt inputs: (1) few-shot task-specific **examples** $\mathcal{E} = \{(l_1, c_1), ..., (l_M, c_M)\}$ that show mappings from instructions to task-specific code plan, and (2) a set of task-agnostic **skills** $\mathcal{Z} = \{z_1, ..., z_N\}$ that the agent can call to compose the plan $c$. Because of context window limits, both $|\mathcal{E}| = M$ and $|\mathcal{Z}| = N$ are fixed. The goal is that the examples and skills are representative enough for the agent to generalize to unseen pairs $(l', c')$, assuming $l'$ reflects the human's intent and the LLM can interpolate from $\mathcal{E}$ and $\mathcal{Z}$ to produce the correct $c'$.

Key to our approach is **dynamically managing the context window**. As the number of examples grows, appending all of them to the prompt is infeasible, and many are irrelevant. For example, when solving "press the button", examples about "open the drawer" add noise and overwhelm the prompt. To address this, we use an external memory module with two vector databases: one for few-shot examples $(l, c)$ indexed by their instructions, and one for skills $z$ indexed by their docstrings. We implement this using ChromaDB and compute embeddings with OpenAI's text-embeddings-3 Neelakantan et al. (2022). For a new instruction $l'$, we retrieve the $K = 10$ most similar examples based on cosine similarity and append them to the prompt. We also retrieve the relevant skill headers from memory and include them in the prompt. Additionally, our framework introduces a simple yet efficient mechanism for guiding the agent with a shared free-form language format: *hints*. Hints allow users to explicitly indicate which previously learned skill may assist with the current task, effectively triggering retrieval from the library of known behaviors. Because our skill library is developed bottom-up, while long-horizon tasks require top-down decomposition, automatic similarity-based retrieval may occasionally overlook a relevant skill, making user-provided hints a simple yet highly effective way to guide the agent toward the correct skills. As the number of skills $|\mathcal{Z}|$ and behaviours $|\mathcal{E}|$ grows, most are irrelevant, and hints provide a way for the user to direct the agent toward the correct subset. If a required sub-behaviour has not yet been learned and cannot be resolved with a hint, the failure signals the user to pause the current skill and teach the missing sub-behaviour first.

With this pipeline, our framework can tackle extremely challenging long-horizon tasks such as "build a house" and "stack a jenga tower", which require more than 20 planning steps (Fig. 2 Phase III left). We further validate our method on widely used benchmarks, including Metaworld and Franka Kitchen, covering over 20 task variations where the framework consistently solves all tasks (Fig. 2 Phase III middle). Finally, we deploy the framework in real-world settings on a franka FR3, successfully completing diverse tasks such as building a house, writing "ICLR", and generalizing to the novel task "build a temple" (Fig. 2 Phase III right). It is worth noting that although we present the learning pipeline in three phases, **the process is not strictly linear**. Whenever the agent cannot handle the current task, the user can roll back to the skill learning or extension phase and iterate until the agent meets the requirements. The overall skill learning process is summarized in Appendix A.

# 3 EXPERIMENTS

## 3.1 EXPERIMENT SETUP

Our work focuses on tabletop long-horizon manipulation tasks. For fair comparison, we include several code and language generation baselines in our experiments (More setup details in Appendix B). Recent work, Voyager Wang et al. (2023), explores skill learning with LLM-based automatic feedback, but it is designed for the Java-based game "Minecraft" with simple action primitives. Adapting it for Python-based robotic manipulation is costly and impractical. Thus, we design a variation of our framework that mirrors their idea, where an LLM provides feedback instead of humans for final task plan generation. We also include a w/o memory version of our framework for ablation that simulates the human-in-the-loop updates at the prompt level Arenas et al. (2024):

**Code as Policy (CaP)** Liang et al. (2023): A representative baseline for LLM-based open-loop code generation without correction or retrieval.

**LoHoRavens (GPT-4o)** Zhang et al. (2023b): A language-generation baseline with explicit LLM feedback, using a pretrained RL policy CLIPort for execution.

**DAHLIA** Meng et al. (2025b): A state-of-the-art code generation framework with LLM-based closed-loop control and incremental examples for in-context learning.

**LYRA (Ours) w/ LLM feedback**: A variation of our framework that relies on LLM feedback, where RGB-D scenes before and after execution are given to LLM to determine task success and provide feedback. This variation has access to our well-learned memory database.

**LYRA (Ours) w/o memory**: A variant without the retrieval module, where 25 out of 49 learned skills and 25 out of 86 explored examples in Ravens are randomly sampled and appended until the context window is filled. For Franka Kitchen and MetaWorld, we sample half of the learned skills and examples at random to evaluate how irrelevant data affects feedback efficiency.

For simulation, we build on the PyBullet-based Ravens benchmark Zeng et al. (2021); Zhang et al. (2023b), where a UE5 robot manipulates multiple tabletop objects. Following prior work Zhang et al. (2023b), we allow LLMs to set up tasks so scenarios can be modified by user intention. To test scalability, we also evaluate on Franka Kitchen Gupta et al. (2020) (long-horizon tasks with a Franka Panda in a kitchen scene) and Metaworld Yu et al. (2020) (Sawyer with diverse tabletop tasks). In the real world, we deploy the agents on a Franka FR3 and validate performance on challenging long-horizon tasks. As our simulation tasks rely on privileged information (e.g., object pose and size), we adopt state-of-the-art perception foundation models Ren et al. (2024); Wen et al. (2024) to obtain open-world object data. Details of the real-world deployment are provided in Appendix C.

## 3.2 SKILL LEARNING WITH HUMAN-IN-THE-LOOP

To analyze how human-in-the-loop skill learning improves performance, we evaluate all models on the customized Ravens with six long-horizon tasks for a case study (Fig. 3(a), setup in Appendix B). For models with feedback, we report task plan results after up to five iterations. For CaP with open-loop control, we report the outcome of each fresh generation. All models are tested with 20 attempts, and we report the average success rate (SR) for comparison. As shown in the figure, CaP performs well only on tasks included in the prompt examples. For tasks outside the prompts, such as "build a 4 x 3 x 2 pyramid" in 3 of 5 unseen structure building tasks, it fails to generalize, achieving only a 0.45 SR. Compared with open-loop CaP, LoHoRavens and DAHLIA improve performance by using LLMs for closed-loop feedback. However, relying solely on LLM feedback can cause the agent to get stuck in long-horizon planning. Our framework variation, LYRA w/ LLM feedback, further underscores this issue. Although the agent has access to all learned skills and examples in the database, the LLM's limited reasoning makes it uncertain about which skills to apply or what the agent can actually perform. As a result, this causes a drop in performance, and the agent may get stuck on extremely long-horizon tasks, with an average SR of only 0.77. Moreover, LYRA w/o memory simulates methods that extend data directly in the prompt. With well-learned skills and examples, this variant represents the upper bound of such methods. As shown in the figure, its performance drops substantially due to the limited prompt length, and unrelated examples in the

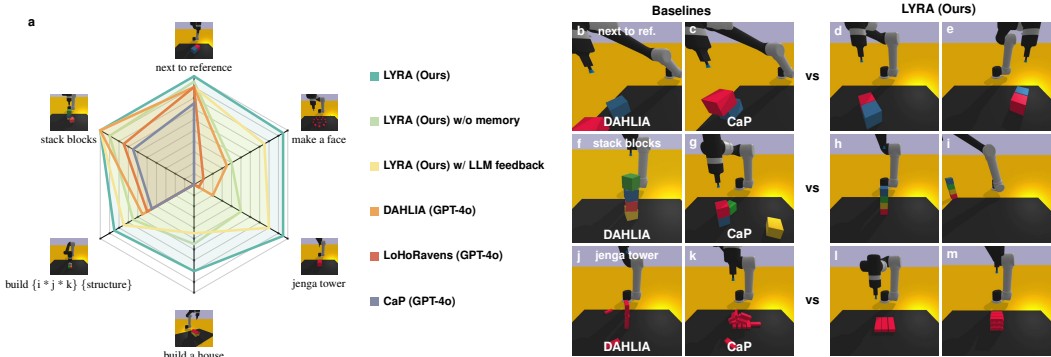

Figure 3: Empirical analysis of skill learning. **(a)** Case study: Baseline comparison on Ravens long-horizon tasks, reported as the average success rate over 20 attempts. **(b)**–**(m)** Snapshots comparison: why human-in-the-loop is essential for skill learning and how it improves performance.

prompt further interfere with reasoning, resulting in an average SR of 0.66. By comparison, our framework achieves a high SR of 0.93, with up to **27% SR improvement** over the variant models.

Relying only on success rate is limited, as it cannot capture alignment with user preferences. A key contribution of our approach is representing human preferences from language corrections in a form that can be preserved and reused through skill inheritance. Snapshots in Fig. 3(b)–(m) show how user feedback improves task performance beyond what SR reflects. For example, in (b) vs. (d), (e), and (f) vs. (h), (i), the baseline places blocks at the target positions, but their poses and distances do not match user expectations. Unlike LLM-only feedback, which often labels sub-optimal results as successful, user interventions guide the agent to acquire precise skills (e.g., stacking corner-to-corner or aligning blocks with fixed spacing) that lead to more reliable and preference-aligned outcomes. Fig. 3(j), (k) vs. (l), (m) further highlights the reliability of human-in-the-loop skill learning. In this example case, we observe **false positives** from LLM evaluation, where incomplete tasks are incorrectly marked as successful. For instance, in Fig. 3(j), DAHLIA stacks long blocks in the same direction for a Jenga tower. Although clearly incorrect to a human, the LLM still labels it as correct, raising doubts about evaluation reliability. In contrast, our framework reuses the "build jenga layer" skill, which was learned earlier with human guidance in a simpler scene, to stack a stable Jenga tower. Results for all Ravens tasks refer to Appendix D and supplementary video.

Skill-oriented human-in-the-loop feedback is more efficient than task-specific flat code feedback from LLMs. Fig. 4(a) shows the average number of corrections (NoC) across Franka Kitchen (4 subtasks) and MetaWorld (20 tasks). A lower NoC means the framework achieves success and aligns with user expectations more efficiently. Both DAHLIA and our LYRA variant w/o memory require more feedback iterations (4.75 and 5.00 in Franka Kitchen) compared to our LYRA framework (2.75). Although LYRA w/o memory has access to learned skills, the presence of irrelevant information in the prompt hinders reasoning and leads to more feedback iterations. In MetaWorld, DAHLIA struggles to complete several tasks within 10 feedback rounds (e.g., hammer), showing the difficulty of fixing low-level execution errors through LLM feedback alone. By comparison, our LYRA framework achieves the fewest corrections in both benchmarks (2.75 in Franka Kitchen and 2.55 in MetaWorld), yielding an average **42% efficiency improvement** over baselines. This gain comes not only from learning skills with human guidance, but also from our **hint** mechanism. Hints let users guide the agent when RAG retrieval fails or when irrelevant data is retrieved. By clarifying what the agent can do and which skills to apply, hints reduce unnecessary corrections compared to LLM feedback. Fig. 4(b)–(i) shows snapshots of our framework's performance on both benchmarks (Full results in Appendix D and supplementary video).

## 3.3 CAN YOU BUILD A HOUSE?

To demonstrate how user-guided lifelong learning expands skill capabilities and enables challenging long-horizon tasks, we present the case study "build a house". As shown in Fig. 5(a), the task required 12 skills developed bottom-up from core primitives (orange), including 7 learned specifically for house building (light green) and others inherited from prior tasks (yellow). Starting from the

**a** Average number of corrections (NoC) (↓)

| Framework | Franka Kitchen (4) | MetaWorld (MT20) |
|---|---|---|
| DAHLIA | 4.75 | $> 10^{\dagger}$ |
| LYRA (Ours) w/o memory | 5.00 | 4.15 |
| LYRA (Ours) | 2.75 | 2.55 |

[†]Some tasks may not be completed within 10 feedback rounds. For example, in the hammer task, an incorrect grasp may cause the hammer to drop, which LLM-based closed-loop feedback systems may struggle to correct, often leading to task failure.

Figure 4: Scalability and feedback efficiency validation. **(a)** Average number of corrections for LLM-based flat code generation versus our human-in-the-loop skill code generation. **(b)**–**(i)** Snapshots from Franka Kitchen and MetaWorld; additional results are provided in Appendix D.

chaotic initial scene (Fig. 5(b)), the task can be decomposed top-down into three subtasks: organizing the scene (Fig. 5(c)), building the base (Fig. 5(d)), and constructing the roof (Fig. 5(e)). Unlike LLM-only closed-loop methods that directly generate flat code without awareness of required skills, our framework acquires intermediate skills such as stacking blocks corner-to-corner or aligning them with precise distance and orientation, which can be further extended to build cubes and more complex structures. Through lifelong learning, users introduce increasingly complex tasks, and the agent expands its capability by reusing existing skills, either nesting them within new ones or modularizing them to develop extended functionality. Both learned skills and examples are stored in an external database, enabling relevant information to be retrieved and appended to the prompt. This prevents catastrophic forgetting and gradually builds a complex skill tree for task planning. This integration of **top-down planning** and **bottom-up skill learning** relies on human verifiers' global awareness of which skills are necessary, a capability current LLM verifiers lack, as they tend to decompose tasks in a single direction and often get stuck in sub-optimal solutions. To the best of our knowledge, our framework is the first to successfully accomplish the "build a house" task, with full implementation details provided in Appendix E.

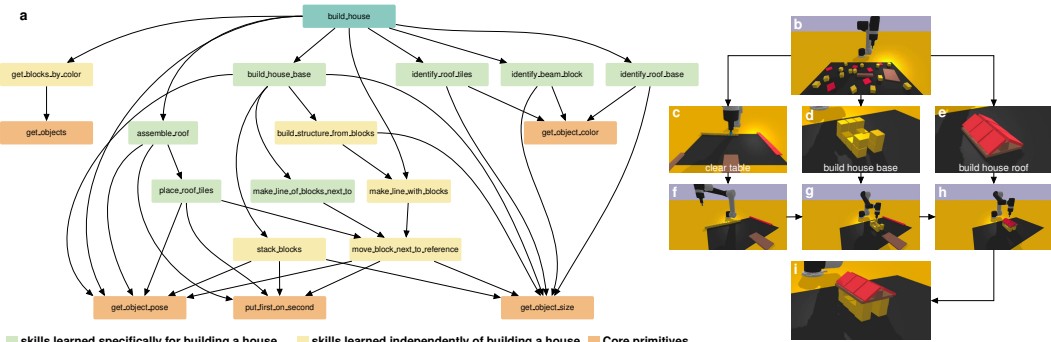

Figure 5: **(a)** Skill tree of build a house. **(b)**-**(i)** Snapshots of task build a house, where **(c)**-**(e)** show sub-skills acquired from bottom-up lifelong skill expansion, and **(f)**-**(h)** show how they are deployed and reused in the task-specific plan generation. Full code implementation in Appendix E.

## 3.4 REAL-WORLD PERFORMANCE

The skills learned through our pipeline are embodiment-agnostic and can be deployed on heterogeneous robot arms, enabling smooth transfer to real-world settings. We provide ROS2-based control APIs for the Franka FR3 with consistent naming and variables to match the simulation environment. As shown in Fig. 6(a)–(d), our framework allows the agent to robustly perform challenging long-horizon real-world tasks, including building a house, stacking a Jenga tower, and writing "ICLR". These tasks can last over 12 minutes, highlighting the framework's effectiveness in real-world deployment. We also test generalization to unseen tasks, such as "construct a temple". Within five rounds of user guidance, the agent reorganizes the scene, builds the temple base, layers, and head by retrieving and combining existing skills, and successfully completes the task. Full results, including "make a human face" and comparisons with baseline models in real-world settings, are provided in Appendix F and supplementary video.

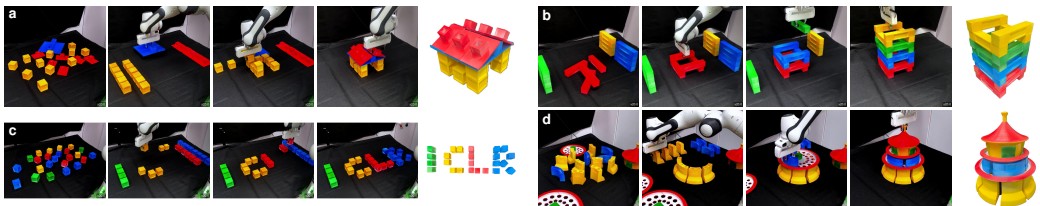

Figure 6: Real-world demonstrations. Snapshots of **(a)** building a house, **(b)** stacking a Jenga tower, **(c)** writing "ICLR", and **(d)** constructing a temple. Full real-world results refer to Appendix F.

## 4 RELATED WORKS

**Code generation in embodied AI.** LLM-based code generation has shown promise by directly translating human intentions into executable code Zhang et al. (2025); Mu et al. (2024). In the game world domain, Voyager Wang et al. (2023) combines an automatic curriculum with a self-verification pipeline, dynamically updating prompts through an extensible skill library, and achieves state-of-the-art performance in Minecraft. In robotics, CaP Liang et al. (2023) is an early approach that lets an LLM generate Python calls to perception and control APIs, enabling robots to grasp objects or navigate from a natural language command. RoboScript Chen et al. (2024) extends this idea by integrating a full ROS pipeline that connects object detection, grasping, and motion planning. However, open-loop control limits the applicability of these methods to complex long-horizon tasks. With recent advances in LLM reasoning, some works now use LLMs as evaluators to provide task-level feedback Zhang et al. (2023b); Meng et al. (2025b); Zhi et al. (2025). However, these approaches neglect human preferences and often fail on extremely long-horizon tasks, leaving the role of humans as verifiers in robotic code generation unexplored.

**Robotic manipulation with language feedback.** Earlier work in interactive imitation learning shows that human feedback can quickly correct trajectory-level errors Co-Reyes et al. (2018); Chisari et al. (2022); Cui et al. (2023); Lynch et al. (2023); Liu et al. (2023). However, these methods usually rely on narrowly defined correction types and fail on out-of-distribution tasks. More recent language generation approaches use LLMs for task decomposition and feedback, while relying on pretrained policies for primitive execution Zhang et al. (2023b); Huang et al. (2023); Guo et al. (2024). These approaches often suffer from environmental perturbations and the limited performance of pretrained policies. Some works also explore humans as verifiers who patch planning mistakes. For example, LMPC Liang et al. (2024) fine-tunes a code-writing model with teacher chat corrections to reduce the number of attempts, but it only generalizes to the taught task and requires large interactive training datasets. PromptBook Arenas et al. (2024) leverages human corrections to refine prompts, achieving robust performance in pick-and-place tasks. However, its effectiveness is limited to the trained scenarios, and it may suffer from catastrophic forgetting when prompts are overwritten. Overall, most methods integrate feedback at the data or prompt level, which makes them prone to catastrophic forgetting in long-horizon task code generation.

**Large-scale robotic foundation models.** VLA models Ma et al. (2024) seek to learn end-to-end mappings from language and vision modalities to robot action outputs. Typical examples include RT-H Belkhale et al. (2024), OpenVLA Kim et al. (2024; 2025), $\pi_{0.5}$ Black et al. (2024); Intelligence et al. (2025), Gemini-Robotics Team et al. (2025) and GR00T N1 Bjorck et al. (2025) etc. However, such models require massive expert demonstrations and computational resources for pre-training. A few studies try to boost data coverage with RAG techniques that pull extra non-robot demonstrations from an external database Ju et al. (2024); Xu et al. (2024); Kuang et al. (2024), but the retrieved data often fail to stay aligned with the robot's current view, state, and goal. Our framework does not try to outcompute these giants; instead, it offers a different path by producing generalized reusable code skills that can be recombined on the fly, giving robust behaviour in unstructured environments.

## 5 CONCLUSION

We presented a human-in-the-loop lifelong skill learning framework that encodes user feedback into reusable and extendable skills, enabling agents to preserve knowledge, extend capabilities, and solve challenging long-horizon tasks such as building a house. Unlike prior closed-loop methods, our ap-

proach integrates skill inheritance, external memory, and a hint-guided retrieval mechanism to align learning with human preferences. Experiments across Ravens, Franka Kitchen, and MetaWorld benchmarks, as well as real-world settings, demonstrate strong performance, achieving a 0.93 success rate (up to 27% higher than baselines) and a 42% efficiency improvement in correction rounds. Our framework robustly handles long-horizon tasks in both simulation and reality, including building a house, stacking a Jenga tower, and generalizing to constructing a temple. We acknowledge the importance of reducing human effort for practical deployment. Having established the viability of skill-based, lifelong knowledge representation in code generation, we see a natural path toward incorporating human–LLM collaborative feedback. This direction offers a promising opportunity to further improve scalability while retaining the core advantages of our skill-centric framework.

## ETHICS STATEMENT

We confirm that this work follows the ICLR Code of Ethics. Our study does not involve human subjects, private or sensitive data, or potentially harmful applications, and all datasets and methods are used in accordance with ethical research practices. We are committed to fairness, transparency, and integrity in both the development and dissemination of our research.

## LLM USAGE STATEMENT

In this study, we employ OpenAI GPT-4o as the embodied agent for robot skill code generation. OpenAI ChatGPT was used solely for sentence-level polishing and grammar correction. All polished content was carefully proofread and verified by the authors for accuracy.

## REPRODUCIBILITY STATEMENT

An anonymous code repository is provided as the supplementary material. To support reproducibility, we include a pre-built skill memory ("trained") that allows readers to replicate our results. This code and data will also be open-sourced upon acceptance.

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

# Appendices

## CONTENTS

## A    PSEUDO CODE OF ALGORITHM

---

**Algorithm 1** Interactive skill learning

---

1: Initialize Skill Library $\mathcal{Z}_0$ and Examples $\mathcal{E}_0$
2: $z \leftarrow$ SkillParser(skill_description)      ▷ Choose current skill to learn
3: **while** True **do**
4:     $l \leftarrow$ task_description      ▷ Provide task instruction
5:     $s_0 \leftarrow$ TaskSetup(initial_state_description)      ▷ Set up the environment
6:     correction $\leftarrow \varnothing$      ▷ Set initial correction
7:     $c \leftarrow \varnothing$      ▷ Set initial task-specific code
8:     **while** True **do**
9:        examples $\leftarrow (l'_i, c'_i)_{i=1,\ldots,K} \in \mathcal{E}$      ▷ Retrieval based on $l$ and correction
10:       $c, z \leftarrow$ Agent($l$, $c$, $z$, correction, examples)
11:       $s_T \leftarrow$ Rollout($c$)      ▷ Roll out policy code
12:       **if** $s_T$ is aligned with $l$ **then**
13:          $\mathcal{E} = \mathcal{E} \cup (l, c)$      ▷ Add example to library
14:          **break**
15:       **end if**
16:       update correction based on $s_T$
17:     **end while**
18: **end while**
19: Update $z$ in $\mathcal{Z}$      ▷ Update skill library

---

# B  FRAMEWORK SETUP

## INTERACTIVE TERMINAL UI

To facilitate smooth and efficient interaction between the human user and the LLM-driven agent, we designed an interactive terminal-based user interface (UI). As illustrated in Fig. S1, the interface is divided into two components. The left panel displays the real-time rendering stream from the underlying simulation engine (e.g., PyBullet or MuJoCo). The right panel allows the user to communicate with the pipeline agent via keyboard input, either by selecting predefined options (such as learning a new skill or re-attempting previous tasks) or by providing free-form textual instructions or feedback.

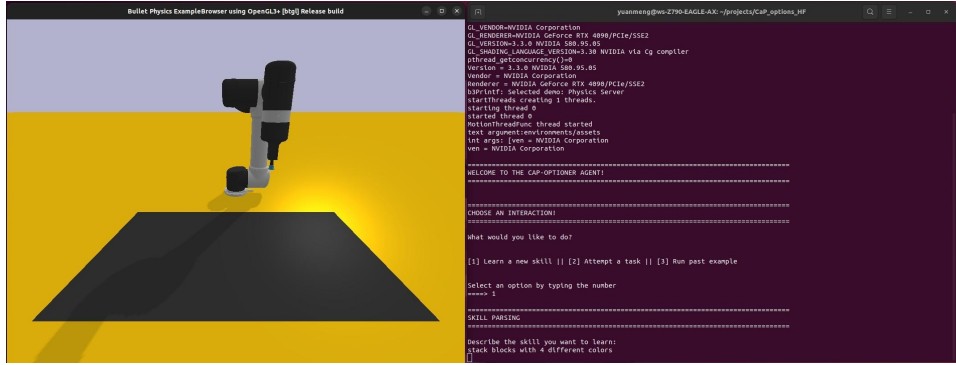

Figure S1: Snapshot of interactive terminal UI

## FRAMEWORK PROMPTING

Our framework and the baseline code generation methods rely on dynamic prompting for in-context learning. Table S1 lists the core primitives used in our framework, and the corresponding prompts are shown below.

Table S1: List of the core-primitives for our agent to build on

| Name | Function |
|------|----------|
| get_objects | This function gets all objects in the environment. The agent can retrieve specific properties of these objects with the functions below. |
| get_object_color | Returns the color of the block. |
| get_object_size | Returns the size of the block. |
| get_object_pose | Returns the pose of the block, given as a 3-dimensional position vector, and a 4 dimensional quaternion rotation. |
| get_bbox | Returns the axis-aligned bounding box of an object, to simplify collision queries. |
| put_first_on_second | The main pick-and-place primitive. It picks up an object at the specified Pose, lifts it vertically to a specified height, moves along the x-y plane to a point directly above the place Pose, then moves it down until it detects contact. |
| move_end_effector_to | Moves the end effector the specified position, and suction gripper rotation. |

We use python format strings for our prompts.

Listing 1: Main Actor prompts

```
actor_system_prompt = f"""
You write python code to control a robotic arm in a simulated environment, building on an existing API.

You will be given:
- a task for the robotic agent to solve
- api functions you may use to solve the task
- if available, examples of codes that solve prior similar tasks

You are supposed to write flat code to solve the task, i.e. do not write any functions.
DO NOT make any imports.

Adhere to the following basic types:
{get_core_types_text()}
"""

def actor_prompt(task, few_shot_examples: list[TaskExample], api: list[Skill]):
    return f"""
    {get_few_shot_examples_string(few_shot_examples)}

    {get_skill_string(api)}
    The task is: {task}

    Write flat code to solve the task.
    """

def actor_iteration_prompt(feedback, examples: list[TaskExample] = []):
    return f"""
    Rewrite the previous code to integrate the feedback: {feedback}.
    {get_few_shot_examples_string(examples)}
    Only make changes that take into account this feedback.
    """
```

Listing 2: Skill Learning prompts

```
actor_skill_learning_system_prompt = f"""
You write python code to control a robotic arm in a simulated environment, building on an existing API.
We are trying to learn skills, and are using different tasks to test and effectively learn a specific skill.

You will be given:
- a task for the robotic agent to solve
- the skill you are supposed to use to solve the task

You are supposed to complete the function, as well as flat, task-specific code, as follows:

def given_function(...) -> ...:
    \"\"\" ... \"\"\"
    <function code>

<task-specific code>

For example:
-------------------------------------------------
IN:
task: "put the red block on the green block"
skill:
def put_block_on_other_block(block: TaskObject, otherBlock: TaskObject):
    \"\"\" places the block on top of otherBlock \"\"\"
    pass

OUT:
def put_block_on_other_block(block: TaskObject, otherBlock: TaskObject):
    \"\"\" places the block on top of otherBlock \"\"\"
    put_first_on_second(get_object_pose(block), get_object_pose(otherBlock))

red_block = get_block(color="red")
green_block = get_block(color="green")
put_block_on_other_block(red_block, green_block)
-------------------------------------------------

If the new task requires you to rewrite the function header, you may do so, for example,
to add arguments, or to update the docstring with important usage information.
You should try to preserve the previous functionality though, since the function might
have previously been used to solve other tasks, which should remain solvable after changes.

DO NOT make any imports.
DO NOT write any functions other than the given one.

Adhere to the following basic types:
{get_core_types_text()}

"""

def skill_learning_prompt(
    task,
    few_shot_examples: list[TaskExample],
    skill: Skill,
    other_useful_skills: list[Skill],
):
```

```
    return f"""
    The task is: {task}
    The function you are supposed to implement is:

    {str(skill)}

    ---------------------------------------------------------------
    {get_few_shot_examples_string(few_shot_examples)}

    The following skills may be useful in your implementation:
    {"\n\n".join([skill.description for skill in other_useful_skills])}
    ---------------------------------------------------------------
    Implement the function and solve the task, while trying to ensure that prior tasks remain solvable.
    """
```

Listing 3: Task Setup prompts

```
task_setup_api_string = """
def add_block(
        self,
        env: Environment,
        color=None,
        size: tuple[float, float, float] = (0.04, 0.04, 0.04),
        pose: Pose=None
    ):
\"\"\" adds a block of a given size and color to the environment
If the pose is left unspecified, a random collision-free pose is selected
 \"\"\"

def add_zone(
        self,
        env: Environment,
        color: str,
        scale: float = 1,
        pose: Pose = None
    ):
\"\"\" adds a zone of a given size and color to the environment
If the pose is left unspecified, a random pose in the workspace is selected
 \"\"\"

def add_cylinder(self, env: Environment, color: str = "red", scale: float = 0.5):
\"\"\" adds a cylinder of a given scale and color to the environment \"\"\"
"""

task_setup_system_prompt = f"""
You are writing python code to setup a simulated environment, translating user instructions
into executable code, based on an existing API.

You should adhere to the following types:
{get_core_types_text()}

You may use the following API:
{task_setup_api_string}

EXAMPLES:
#########

task: add 3 red blocks and 3 blue blocks
response:
for _ in range(3):
    self.add_block(env, "red")

for _ in range(3):
    self.add_block(env, "blue")

#########

task: add one big block and 4 blocks that are a quarter of the big blocks side length
response:
self.add_block(env, size=(0.08, 0.08, 0.08))
for _ in range(4):
    self.add_block(env, size=(0.02, 0.02, 0.02))

#########
"""
```

Listing 4: Skill Parser prompts

```
generate_function_header_system_prompt = f"""
We are working in the context of controlling a robotic arm with python code.
The user proposes a certain skill they would like the robot to learn.
To enable this, you are supposed to translate this skill into a python function,
i.e. choose a clear, descriptive name for the function, choose appropriate arguments,
and write a clear, descriptive docstring.

For example:
```

```
USER: "place one block on top of the other"
RESPONSE:
def place_block_on_other_block(block: TaskObject, otherBlock: TaskObject):
    \"\"\" Places one block on top of the other block  \"\"\"
    pass

Do not try to implement the function yet, that happens later.
You should adhere to the following types:
{get_core_types_text()}

The functions don't need Workspace as an argument, since there is only one.
"""

def generate_skill_prompt(prompt, similar_skills: list[Skill]):
    return f"""
    you may use the following function headers as examples of what you are trying to generate:
    {"\n".join([skill.description for skill in similar_skills])}
    -------------------------------------------------
    write a function header for the prompt: {prompt}.
    """

def refine_function_header_prompt(function_code, refinement):
    return f"""
    Your role is to refine an existing python function, for example by adding a function argument or
    changing the name. If the function is implemented (i.e. not just "pass"), you should also alter
    the implementation accordingly, making as little changes and assumptions as possible.
    Revise the following python function according to the user instructions:
    {function_code}
    Refinement prompt:
    {refinement}
    Do not make any assumptions.
    """

class ParsedList(BaseModel):
    parsed_list: list[str]

def parse_hint_to_list_prompt(hint):
    return f"""
    The user provided a list of tasks that are similar to the one you are currently
    trying to solve, in a single string. Retrieve each of the task descriptions from
    this string and return them as a list.
    This is the string: {hint}
    """
```

Listing 5: Core primitives

```
"""
---------------------------------------------------------------------------
the following functions require an initialised environment -
the agent doesn't need to know anything about the environment, only what methods are available to it
we are responsible for properly initialising the environment, and ensuring that the agent has access to it
"""

"""
IMPORTANT
- pybullet can only handle one server at a time, if this is not commented out, this is the environment being
    used
"""

# env = Environment(
#     "/Users/maxfest/vscode/thesis/thesis/environments/assets",
#     disp=True,
#     shared_memory=False,
#     hz=480,
#     record_cfg={
#         "save_video": False,
#         "save_video_path": "${data_dir}/${task}-cap/videos/",
#         "add_text": True,
#         "add_task_text": True,
#         "fps": 20,
#         "video_height": 640,
#         "video_width": 720,
#     },
# )
# from tasks.tasks.place_blocks import Place5Blocks

# task = Place5Blocks()

# env.set_task(task)
# env.reset()
"""---------------------------------------------------------------------------"""

__all__ = [
    "get_objects",
    "get_object_size",
    "get_object_pose",
    "get_object_color",
    "get_end_effector_pose",
    "put_first_on_second",
    "move_end_effector_to",
    "get_bbox",
```

```
972        "get_point_at_distance_and_rotation_from_point",
973    ]
974    def get_objects() -> list[TaskObject]:
975        """gets all objects in the environment"""
976        return env.task.taskObjects
977    def get_object_size(task_object: TaskObject) -> tuple[float, float, float]:
978        """Returns the size of the given TaskObject as a tuple (width, depth, height)."""
979        return task_object.size
980    def get_object_pose(obj: TaskObject) -> Pose:
981        """returns the pose (Point3d, Rotation) of a given object in the environment."""
982        return _from_pybullet_pose(env.get_object_pose(obj.id))
983    def get_object_color(task_object: TaskObject) -> str:
984        return task_object.color
985    def get_end_effector_pose() -> Pose:
986        """gets the current pose of the end effector"""
987        return _from_pybullet_pose(env.get_ee_pose())
988    def put_first_on_second(pickPose: Pose, placePose: Pose):
989        """
990        This is the main pick-and-place primitive.
991        It allows you to pick up the TaskObject at 'pickPose', and place it at the Pose specified by 'placePose'.
992        If 'placePose' is occupied, it places the object on top of 'placePose'.
993        """
994        return env.step(
995            action={
996                "pose0": _to_pybullet_pose(pickPose),
997                "pose1": _to_pybullet_pose(placePose),
998            }
999        )
1000   def move_end_effector_to(pose: Pose, speed=0.001):
1001       """moves the end effector from its current Pose to a given new Pose"""
1002       env.movep(_to_pybullet_pose(pose), speed=speed)
1003   def get_bbox(obj: TaskObject) -> AABBBoundingBox:
1004       """gets the axis-aligned bounding box of an object - this is useful primarily for collision detection"""
       aabb_min, aabb_max = env.get_bounding_box(obj.id)
       return AABBBoundingBox(Point3D.from_xyz(aabb_min), Point3D.from_xyz(aabb_max))
   def get_point_at_distance_and_rotation_from_point(
       point: Point3D, rotation: Rotation, distance: float, direction=np.array([1, 0, 0])
   ) -> Point3D:
       """compute a point that is at a specific 'distance' from 'point', at a specified 'rotation'
       The direction specifies the base direction in which to apply the rotation.
       This is useful for placing objects relative to other objects.
       """
       rotated_direction = rotation.apply(direction)
       new_point = point.np_vec + distance * rotated_direction
       return Point3D.from_xyz(new_point)
```

BASELINE MODEL SETUP

To ensure a fair comparison, we leverage state-of-the-art open-source code and language generation repositories. We adapt the baseline implementations and provide wrappers for direct integration with our APIs:

- **CaP**: `https://github.com/google-research/google-research/tree/master/code_as_policies`

- **LoHoRavens**: `https://github.com/Shengqiang-Zhang/LoHo-Ravens`

- **DAHLIA**: `https://github.com/Ghiara/DAHLIA`

We also consider Voyager Wang et al. (2023), which employs LLMs for automatic skill learning and RAG retrieval in Minecraft. However, Voyager is built for a Java-based game environment, and adapting it to our Python-based robotic control is impractical. To enable comparison, we design a variation of our framework where LLMs provide feedback (**LYRA (Ours) w/ LLM feedback**), keeping close to Voyager's concept. Following prior works Zhang et al. (2023b); Meng et al. (2025b); Wang et al. (2023), the LLM evaluates task success by receiving RGB-D observations before and after execution and providing corrective feedback if needed. In practice, we find that letting the LLM directly discover missing capabilities and learn new skills is difficult, since robotic manipulation involves more complex dynamics and spatial reasoning than Minecraft. Therefore, in this variation, we do not attempt direct skill learning but instead rely on the learned skills and example

databases from our framework. The LLM focuses on reasoning about the current task, providing feedback to adjust the plan, and retrieving relevant items from memory.

For ablation, we also test a version without the retrieval module (**LYRA (Ours) w/o memory**). Here, learned skills and examples are randomly sampled from our databases and appended to the prompt until the context window is filled. This simulates catastrophic forgetting in approaches that only modify prompts with human-in-the-loop Arenas et al. (2024). Specifically, we include 25 out of 49 learned skills and 25 out of 86 examples for Ravens tasks. This setup allows us to study how learned skills contribute when recent works rely solely on examples for in-context learning. Although performance drops on more complex tasks due to limited access to the full skill library, this variation still outperforms baseline models.

Since both the baseline models and our framework rely on few-shot in-context learning for reasoning and task planning, we configure the few-shot examples and/or skills for the baselines as follows:

- **Customized Ravens**: All models, including ours, start with 9 core primitives as initial skills, which are also appended to the baselines for in-context learning. For few-shot adaptation, all baselines are provided with the following example task plans and are expected to generalize to unseen scenarios, either in an open-loop manner or under LLM guidance:
  - stack blocks (4 colors)
  - put the yellow block next to the green block
  - build a $\{2 \times 2 \times 2\}$ cube with 8 blocks
  - build a $\{3 \times 2 \times 2\}$ pyramid with 8 blocks
  - put the red block in the middle of the workspace
  - rotate the blue block by 45 degrees
  - move the end effector to the center of the workspace
  - move the smallest block 10cm to the left
  - arrange the blocks around a circle
- **Franka Kitchen & Metaworld**: All models, including ours, start with 9 core primitives as initial skills (shared across both benchmarks via unified API wrappers). For few-shot adaptation, baselines are given the following example task plans and are expected to generalize to new scenarios:
  - reach a goal
  - push a puck to a goal
  - pick and place a puck in a goal
  - open a door (revolute joint)
  - close a window (slide joint)

RAVEN TASKS SETUP

In the Experiment section, we compare the success rates across 6 customized Ravens tasks as a case study. The tasks are set up as follows:

- **Place block next to reference**: A scene with two large blocks (red and blue). The agent must place the red block next to the blue one without collision. Alignment details (e.g., corner-to-corner) are not required for success. This task helps the agent acquire foundational spatial skills.

- **Stack blocks**: A scene with four medium-sized colored blocks. The agent must stack all blocks into a tower. Precise alignment is not required for success evaluation; towers with random positions or orientations are also considered successful. This provides a base skill for more complex behaviors.

- **Build $\{i \times j \times k\}$ structure**: A scene with multiple medium-sized colored blocks. The agent must build structures as instructed by the user. Two example plans are provided in the prompt: (1) a $\{2 \times 2 \times 2\}$ cube and (2) a $\{3 \times 2 \times 2\}$ pyramid. The agent must also generalize to unseen tasks: (3) a $\{4 \times 3 \times 3\}$ pyramid, (4) a $\{1 \times 3 \times 3\}$ wall, and (5) a $\{4 \times 4 \times 1\}$ base.

- **Make a human face**: A scene with several medium-sized colored blocks and one rectangular block. The agent must build a circle and use two blocks plus the rectangle to form facial features inside. Orientation details are not required for success.

- **Build a Jenga tower**: A scene with multiple rectangular blocks of uniform or mixed colors. The agent must stack them in alternating orientations to construct a stable Jenga tower.

- **Build a house**: A scene with blocks of various colors and shapes. The agent must use yellow blocks for the house base, brown blocks for the roof base, and red plates for the tiles.

## C  REAL-WORLD EXPERIMENT SETUP

### HARDWARE SETTING

In the real-world demonstrations, we employ a Franka FR3 manipulator paired with an Intel RealSense D435i depth camera mounted at the table edge to evaluate the proposed framework. As illustrated in Fig. S2, the camera is positioned approximately 1 m in front of the robot, closely mirroring the configuration used in simulation. The robot's base coordinate system, $O_{base}$, is defined at the center of joint 0, with the x-axis directed toward the table and the z-axis aligned vertically upward from the table surface. For object manipulation, the robot is equipped with a Franka two-finger gripper, in contrast to the simulation benchmark, which employs a suction gripper for pick-and-place tasks. To ensure seamless deployment across embodiments and gripper types, we maintain consistent primitive APIs between the real-world and simulated settings, enabling skills developed in simulation to transfer directly to the physical robot.

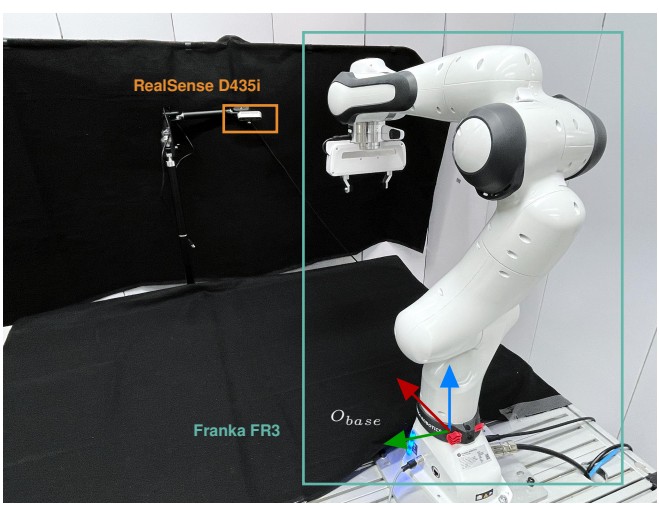

Figure S2: Real-world hardware setup. A Franka FR3 serves as the agent embodiment, with an Intel RealSense D435i placed at the table edge to capture privileged scene information.

In our real-world setup, the calibrated extrinsic transformation between the camera and the robot base frame is obtained as:

$$T_{\text{base}}^{\text{cam}} = \begin{bmatrix} 6.12323400e-17 & 7.37277337e-01 & -6.75590208e-01 & 1.06 \\ 1.00000000e+00 & -4.51452165e-17 & 4.13679693e-17 & 0.16 \\ -0.00000000e+00 & -6.75590208e-01 & -7.37277337e-01 & 0.61 \\ 0. & 0. & 0. & 1.00 \end{bmatrix} \quad (1)$$

### PERCEPTION

In simulation, privileged object information (such as type, size, pose, and color) can be directly accessed through predefined APIs, e.g., get_objects, get_object_size, and get_object_pose. In contrast, acquiring such information in the real world is more challenging. To address this, we employ recent vision foundation models to perceive and parse the task scene. For open-world object pose estimation, we use Grounded SAM 2 Ren et al. (2024) to detect target object masks and their bounding boxes (with SAM 2.1.hiera.large and GroundingDINO.swint.ogc). The object center is estimated by computing the mean of all mask pixels and projecting it back into 3D space. For orientation estimation, we adopt FoundationPose Wen et al. (2024) in combination with object CAD models to recover the object's coordinate frame, particularly the x- and y-axis directions. By comparing the angle differences between the object and robot base coordinate frames, we determine the object's rotation pose.

DEPLOYMENT

We utilize the official Franka ROS2 library on Ubuntu 22.04 to control the Franka FR3 manipulator. The action space of our framework is defined in a 6-dimensional vector space, representing end-effector motions composed of linear displacements along the x, y, and z axes, as well as rotational changes in roll, pitch, and yaw. By default, Franka ROS2 relies on OMPL-based motion planning. To better align with our simulation environment and enable more intuitive trajectory control, we extend this setup by customizing a Cartesian linear motion planning instance, which allows for straightforward and precise linear path execution. To ensure reliable operation in the real-world hardware setup, we carefully configure the planning parameters, including detection tolerances, velocity and acceleration scaling factors, and orientation/pose constraints. The complete parameter configuration for our motion planning setup is summarized in Table S2, which provides the exact values adopted in our experiments.

Table S2: Hardware parameter setup

| Parameters | Values |
|---|---|
| RealSense x-axis detection tolarance | 0.01 [m] |
| RealSense y-axis detection tolarance | 0.01 [m] |
| RealSense z-axis detection tolarance | 0.01 [m] |
| Feasible factor of cartesian linear planning | 0.9 |
| Max velocity scaling factor | 0.1 |
| Max acceleration scaling factor | 0.1 |
| Absolute x-axis orientation tolerance | 0.02 |
| Absolute y-axis orientation tolerance | 0.02 |
| Absolute z-axis orientation tolerance | 0.02 |
| Target pose constratint in x-axis | 0.005 [m] |
| Target pose constratint in y-axis | 0.005 [m] |
| Target pose constratint in z-axis | 0.005 [m] |
| Workspace in x-axis | +0.25 [m] - +0.8 [m] |
| Workspace in y-axis | -0.55 [m] - +0.3 [m] |
| Workspace in z-axis | +0.01 [m] - +0.65 [m] |

The following is an example prompt for generating a task-specific code plan that can be deployed in the ROS2 environment.

Listing 6: Demo task script structure for ROS2 deployment

```python
from utils.core_types import Workspace
# The task script should be sumamrized as follows to enable a direct deployment in ROS2 environment:

def demo(record_cfg, camera_args):
    # --------------------------------------------------------------------------------
    # Do not modify this part, because all task demo are execute under ROS2 environment node
    rclpy.init()
    executor = MultiThreadedExecutor()
    env = RealWorldEnvironment(record_cfg=rec_cfg, camera_args=camera_args)
    executor.add_node(env)
    executor_thread = threading.Thread(target=executor.spin, daemon=True)
    executor_thread.start()
    env.reset()
    # --------------------------------------------------------------------------------
    try:

        # ... You can implement your task specific code at here ...
        # ... flat code ...
        # ... call your skill_function(...) to accomplish the task ...

    # --------------------------------------------------------------------------------
```

```
        # You donot need to touch this part
    except KeyboardInterrupt:
        # env.end_rec()
        print("\n Test interrupted by user")
    except Exception as e:
        # env.end_rec()
        print(f"\n Test failed: {str(e)}")
        import traceback
        traceback.print_exc()
    finally:
        try:
            env.destroy_node()
            # env.end_rec()
        except:
            pass
        executor.shutdown()
        rclpy.shutdown()
    # --------------------------------------------------------------------------------
```

# D SIMULATION PERFORMANCE

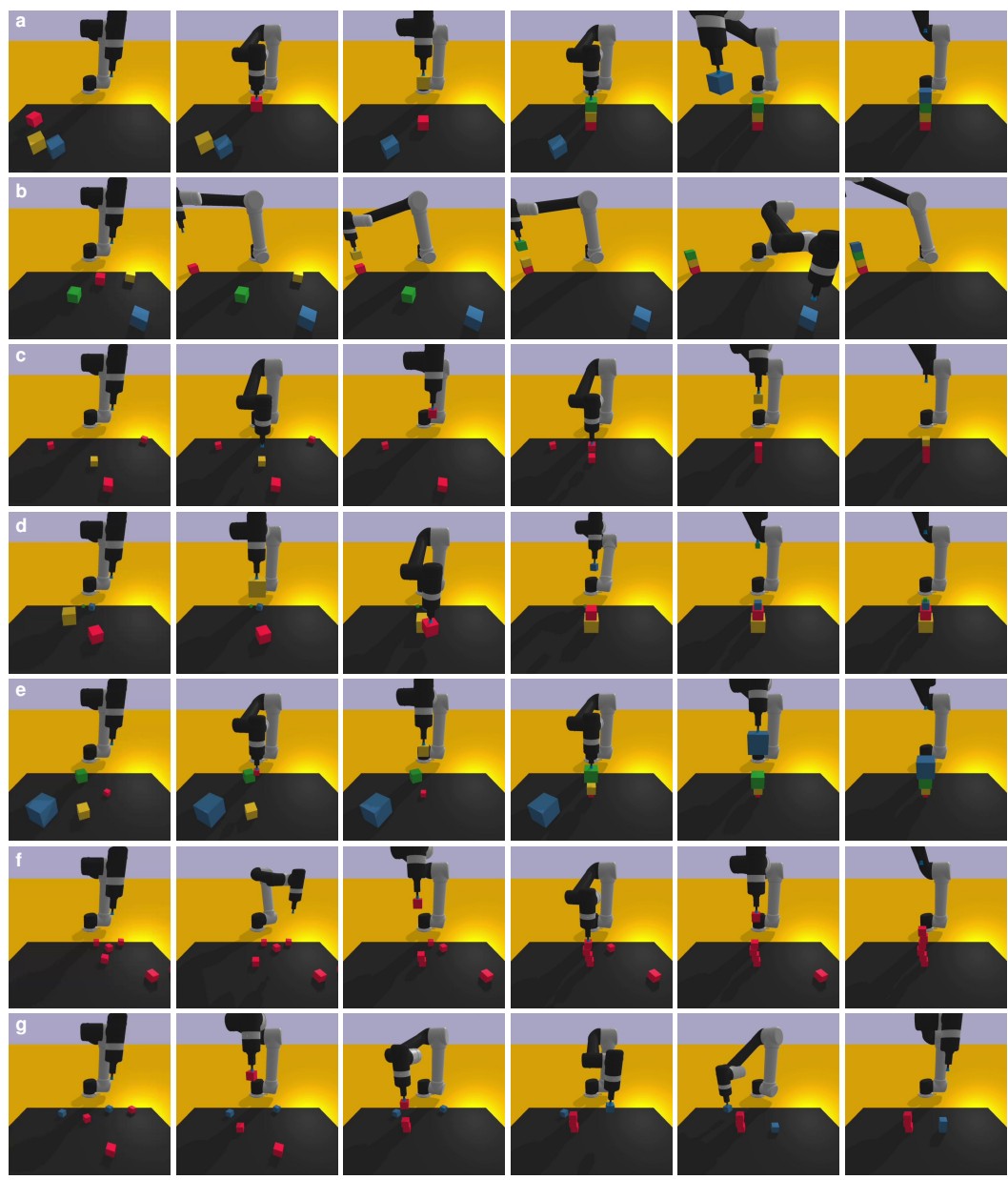

Figure S3: Snapshots of the "stack blocks" task and its variations under user-guided lifelong learning. **(a)** Stack 4 blocks corner-to-corner at the workspace center. **(b)** Stack 4 blocks with a 45° rotation at the back-right table corner. **(c)** Stack blocks by color, with the yellow block on top. **(d)** Stack blocks from largest to smallest. **(e)** Stack blocks from smallest to largest. **(f)** Stack blocks into a zigzag tower. **(g)** Build two towers sorted by color.

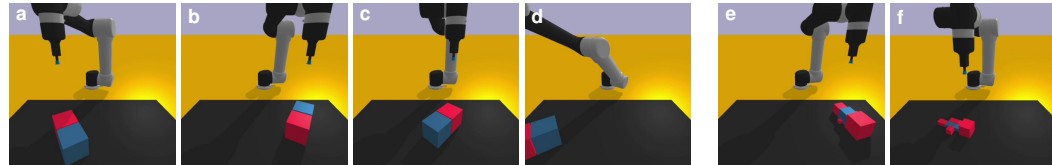

Figure S4: Snapshots of the task "place block next to reference." **(a)**–**(d)** Basic skill that places a red block next to a blue block with a 0.5 cm gap along different axes. **(e)**–**(f)** Extended capability through user-guided lifelong learning, where the agent places red blocks on all sides of the blue block to form a cross.

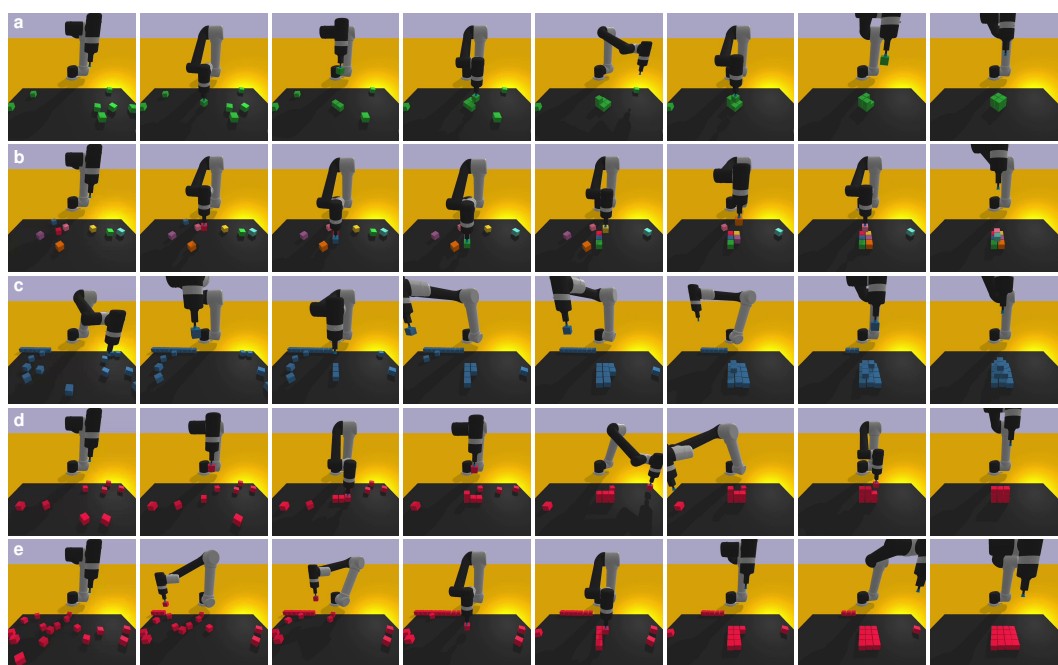

Figure S5: Snapshots of the task "build {i * j * k} {structure}". **(a)** Build a 2 x 2 x 2 cube. **(b)** Build a 3 x 2 x 2 pyramid. **(c)** Build a 4 x 3 x 3 pyramid. **(d)** Build a 1 x 3 x 3 wall. **(e)** Build a 4 x 4 x 1 base.

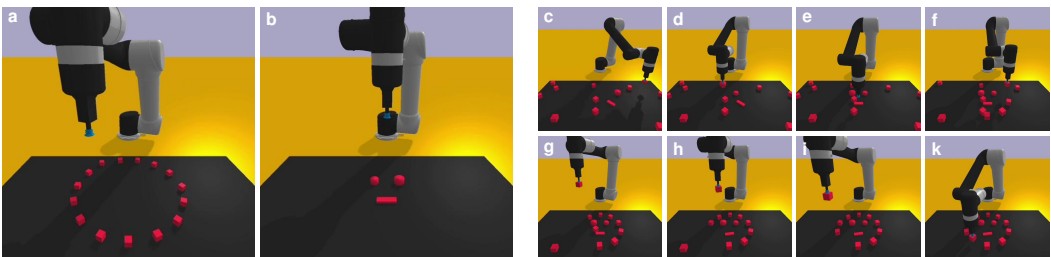

Figure S6: Snapshots of the "make smiley human face" task. **(a)**–**(b)** Basic skills "arrange_in_circle" and "make_face_feature," used to construct the circular outline and the eyes and mouth with blocks. **(c)**–**(k)** Extended capability through user-guided lifelong learning, where the agent recalls and reuses these basic skills to complete the full human face structure.

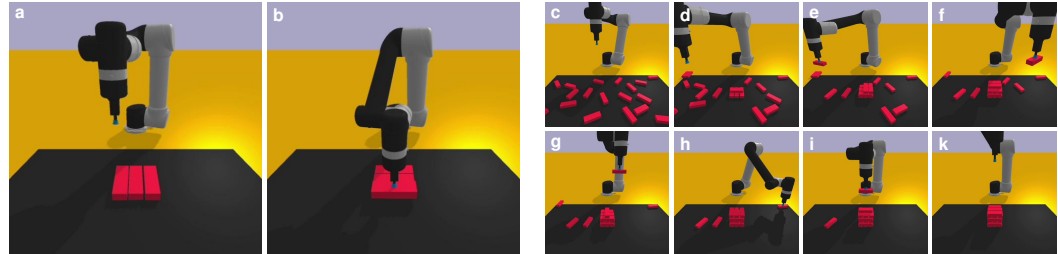

Figure S7: Snapshots of the "build jenga tower" task. **(a)**–**(b)** Basic skill "build_jenga_layer," which constructs a single Jenga layer in the required orientation. **(c)**–**(k)** Extended capability through lifelong learning, where the agent reuses skills to stack layers and complete the Jenga tower.

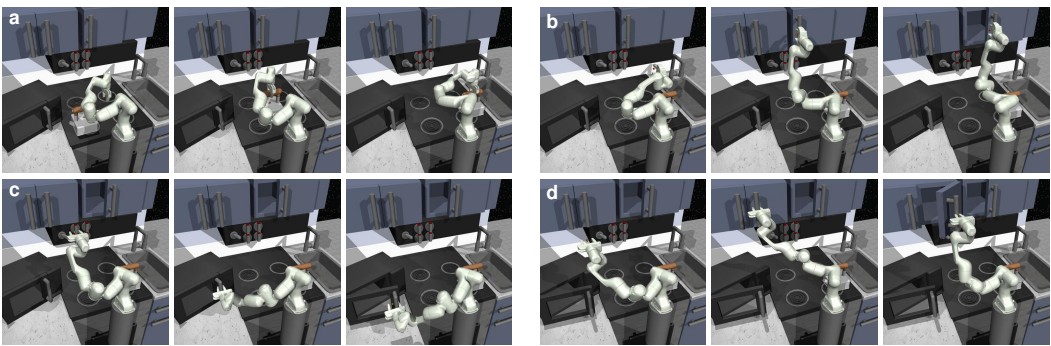

Figure S8: Snapshots of Franka Kitchen. **(a)** pick-place kettle; **(b)** open slide-cabinet; **(c)** open microwave; **(d)** open hinge-cabinet.

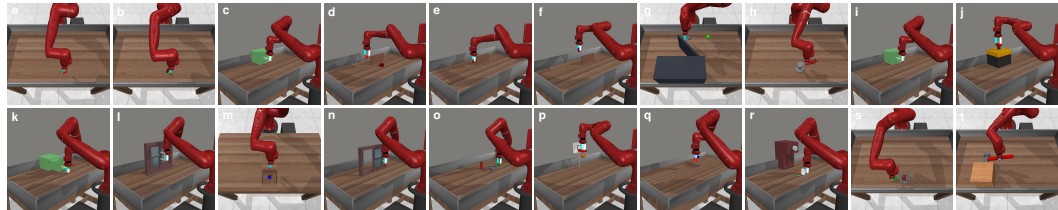

Figure S9: Snapshots of Meta-World. **(a)** Reach; **(b)** Push; **(c)** Pick-place; **(d)** Reach-wall; **(e)** Push-wall; **(f)** Pick-place-wall; **(g)** Door-open; **(h)** Faucet-open; **(i)** Drawer-close; **(j)** Button-press; **(k)** Drawer-open; **(l)** Window-close; **(m)** Sweep-into; **(n)** Window-open; **(o)** Disassembly; **(p)** Basketball; **(q)** Pick-out-of-hole; **(r)** Coffee-pull; **(s)** Assembly; **(t)** Hammer.

Table S3: Number of Corrections (↓) for individual tasks in Franka Kitchen (FK) and MetaWorld (MW). "-" means failed in the task.

| Framework | FK - kettle | FK - slide cabinet | FK - hinge cabinet | FK - microwave | MW - reach | MW - push | MW - pick-place | MW - reach-wall |
|---|---|---|---|---|---|---|---|---|
| DAHLIA | 6 | 3 | 5 | 5 | 0 | 3 | 6 | 7 |
| LYRA (Ours) w/o memory | 4 | 1 | 8 | 7 | 0 | 9 | 6 | 2 |
| LYRA (Ours) | 2 | 2 | 4 | 3 | 0 | 5 | 5 | 3 |
| | MW - push-wall | MW - pick-place-wall | MW - door-open | MW - faucet-open | MW - drawer-close | MW - button-press | MW - drawer-open | MW - window-close |
| DAHLIA | 6 | - | 2 | 3 | 2 | 4 | 3 | 2 |
| LYRA (Ours) w/o memory | 6 | 7 | 1 | 4 | 3 | 4 | 3 | 4 |
| LYRA (Ours) | 4 | 3 | 0 | 3 | 1 | 1 | 4 | 2 |
| | MW - sweep-into | MW - window-open | MW - disassembly | MW - basketball | MW - pick-out-of-hole | MW - coffee-pull | MW - assembly | MW - hammer |
| DAHLIA | 7 | 2 | 4 | 3 | 4 | - | - | - |
| LYRA (Ours) w/o memory | 7 | 3 | 2 | 1 | 5 | 3 | 9 | 4 |
| LYRA (Ours) | 3 | 0 | 1 | 1 | 3 | 2 | 4 | 6 |

# E   BUILD A HOUSE

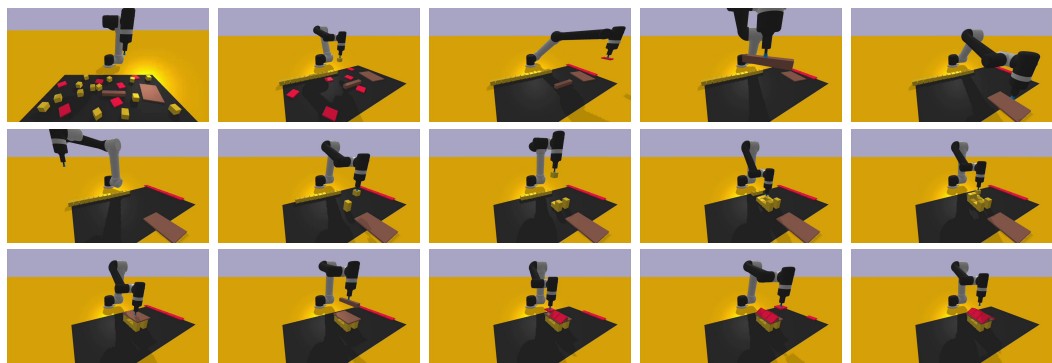

Figure S10: Snapshots of "build a house".

The implementation details of the skill "build a house" are demonstrated as follows:

Listing 7: Function build_house

```python
from utils.core_types import Workspace

def build_house():
    """Builds a house in the middle of the workspace.
    Assumes all the necessary objects are available in the workspace,
    and moves them out of the way before building the house.
    """
    objects = get_blocks_by_color()
    base_blocks = get_blocks_by_color("yellow")
    if len(base_blocks) != 14:
        raise Exception("Not enough blocks to build the house")
    roof_base = identify_roof_base(objects)
    if not roof_base:
        raise Exception("Can't find the roof base")
    roof_beam = identify_beam_block(objects)
    if not roof_beam:
        raise Exception("Can't find the roof beam")
    roof_tiles = identify_roof_tiles(objects)
    if not roof_tiles:
        raise Exception("Can't find the roof tiles")

    # Move objects out of the way
    # Move base blocks to back edge of workspace
    back_left = Workspace.back_left
    make_line_with_blocks(
        base_blocks, Pose(back_left, Rotation.from_euler("z", np.pi / 2))
    )

    # Move roof tiles to right side of workspace
    back_right = Workspace.back_right
    make_line_with_blocks(roof_tiles, Pose(back_right, Rotation.identity()))

    # Move roof beam to front edge of workspace
    front_left = Workspace.front_left
    put_first_on_second(
        get_object_pose(roof_beam), Pose(front_left, Rotation.identity())
    )

    # Move roof base to middle of the front edge
    front_middle = Point3D(Workspace.front_left.x, Workspace.middle.y, 0)
    put_first_on_second(
        get_object_pose(roof_base), Pose(front_middle, Rotation.identity())
    )

    # Build the house
    middle = Workspace.middle
    build_house_base(base_blocks, Pose(middle, Rotation.identity()))

    assemble_roof(roof_base, roof_beam, roof_tiles, Pose(middle, Rotation.identity()))
```

Listing 8: Function build_house_base

```python
def build_house_base(blocks: list[TaskObject], pose: Pose):
    """Constructs the base of a house in the workspace using a list of block
    TaskObjects starting from a specified pose.
    Args:
    blocks (list[TaskObject]): A list of blocks to use in forming the base of a house.
    Each block should have uniform attributes such as size and color.
    startingPose (Pose): The starting position and orientation in the workspace from
    which to begin constructing the base of the house.
```

```
1512
1513      """
1514      block_width = get_object_size(blocks[0])[0]
          startingPose = Pose(
1515          pose.position.translate(Point3D(1.5 * block_width, -block_width, 0)),
              pose.rotation,
1516      )
1517      # make the front left corner of the house
1518      stack_blocks(blocks[0:2], startingPose)
1519      # leave a gap of one block width
1520      next_stack_position = get_point_at_distance_and_rotation_from_point(
              startingPose.position,
1521          startingPose.rotation,
              block_width * 2,
1522          direction=(0, 1, 0),
1523      )
          stack_blocks(blocks[2:4], Pose(next_stack_position, startingPose.rotation))
1524      # make lines towards the back of the workspace from each of the stacks
1525      make_line_of_blocks_next_to(blocks[4:6], blocks[0], "back")
          make_line_of_blocks_next_to(blocks[6:8], blocks[3], "back")
1526
1527      # make the back wall of the house
          back_wall_start_pos = get_point_at_distance_and_rotation_from_point(
1528          get_object_pose(blocks[5]).position,
              startingPose.rotation,
1529          block_width + 0.005,
              direction=(-1, 0, 0),
1530      )
1531
          build_structure_from_blocks(
1532          blocks[8:14], (1, 3, 2), Pose(back_wall_start_pos, startingPose.rotation)
          )
1533
1534
```

Listing 9: Function identify_roof_tiles

```
def identify_roof_tiles(objects: list[TaskObject]) -> list[TaskObject]:
    """Identifies and returns a list of TaskObjects that are categorized as roof tiles
    from a given list of objects. A roof tile is characterized by having one dimension smaller
    than 0.02 and being red in color.
    Args:
    - objects (list[TaskObject]): A list of TaskObjects to be analyzed for identification of roof tiles.
    Returns:
    - list[TaskObject]: A list of TaskObjects that are identified as roof tiles,
    based on the specified characteristics.
    """
    roof_tiles = []

    for obj in objects:
        if obj.color.lower() == "red" and any(dim < 0.02 for dim in obj.size):
            roof_tiles.append(obj)
    return roof_tiles
```

Listing 10: Function identify_beam_block

```
def identify_beam_block(blocks: list[TaskObject]) -> TaskObject:
    """Identifies the beam block from a list of blocks.
    A beam block is defined by the following criteria:
    - It must have the color 'brown'.
    - It has one square side, meaning two side lengths must be the same.
    - The third side should be at least 3 times as long as the square sides.
    Args:
    blocks (list[TaskObject]): The list of block objects to be evaluated.
    Returns:
    TaskObject: Returns the TaskObject identified as a beam block.
    If no beam block is found, returns None.
    """

    for block in blocks:
        if block.color != "brown":
            continue
        width, depth, height = sorted(block.size)
        if width == depth and height >= 3 * width:
            return block
    return None
```

Listing 11: Function identify_roof_base

```
def identify_roof_base(objects: list[TaskObject]) -> TaskObject:
    """ Identifies and returns the TaskObject that serves as the base for a roof in a given list of objects.
    A roof base is characterized by being brown in color and by having two dimensions that are at least
    10 times larger than the third dimension.
    Args:
    - objects (list[TaskObject]): A list of TaskObjects to be analyzed for identification of the roof base.
    Returns:
```

```
- TaskObject: The TaskObject identified as the roof base, based on the specified characteristics.
"""
for obj in objects:
    size = obj.size
    if obj.color == 'brown':  # Changed from 'red' to 'brown'
        dimensions = sorted(size)
        if dimensions[0] * 10 <= dimensions[1] and dimensions[0] * 10 <= dimensions[2]:
            return obj
return None
```

### Listing 12: Function get_blocks_by_color

```
def get_blocks_by_color(color: str = None) -> list[TaskObject]:
    """
    Retrieves all block objects in the workspace.
    If a specific color is provided, only blocks of that color are retrieved.
    Args:
    color (str, optional): The color of the blocks to retrieve.
                           If not specified, retrieves all blocks regardless of their color.
    Returns:
    list[TaskObject]: A list of TaskObject instances representing the blocks in the workspace.
    """
    all_objects = get_objects()
    if color:
        return [obj for obj in all_objects if obj.objectType == 'block' and obj.color == color]
    else:
        return [obj for obj in all_objects if obj.objectType == 'block']
```

### Listing 13: Function assemble_roof

```
def assemble_roof(base: TaskObject, roof_beam: TaskObject,
                  roof_tiles: list[TaskObject], overall_pose: Pose):
    """
    Assembles a roof structure using a designated base, a roof beam, and a list of roof tiles,
    starting from a given overall pose.The base acts as the foundation while the roof beam provides
    structural support and the roof tiles are placed on top to complete the structure.
    Args:
    - base (TaskObject): The TaskObject representing the base upon which the roof is built.
    - roof_beam (TaskObject): The TaskObject representing the beam supporting the roof tiles
    between the base and the tiles.
    - roof_tiles (list[TaskObject]): A list of TaskObjects representing the roof tiles to
    be placed on the beam.
    - overall_pose (Pose): The Pose indicating the overall position and orientation for the roof assembly.
    """
    # Place base in the middle of the workspace
    put_first_on_second(get_object_pose(base), overall_pose)
    # Compute the pose for the roof beam
    base_pose = get_object_pose(base)
    beam_pose = Pose(
        Point3D(
            base_pose.position.x,
            base_pose.position.y,
            base_pose.position.z + (base.size[2] / 2) + (roof_beam.size[2] / 2)
        ),
        base_pose.rotation
    )
    # Place beam on the base
    put_first_on_second(get_object_pose(roof_beam), beam_pose)
    # Compute the pose for the roof tiles on top of the beam
    beam_pose = get_object_pose(roof_beam)
    roof_tiles_pose = Pose(
        Point3D(
            beam_pose.position.x,
            beam_pose.position.y,
            beam_pose.position.z + roof_beam.size[2] / 2 + 0.01
        ),
        beam_pose.rotation
    )
    # Place roof tiles
    place_roof_tiles(roof_tiles, roof_tiles_pose)
```

### Listing 14: Function build_structure_from_blocks

```
from utils.core_types import *

def build_structure_from_blocks(
    blocks: list[TaskObject],
    dimensions: tuple[int, int, int],
    pose: Pose,
    gap: float = 0.005,
):
    """
    Assembles a structure using individual block TaskObjects based on the specified dimensions
    and the given pose. The blocks should be positioned to form the desired structure starting
    from the given pose, which specifies the position and orientation of the first block placed.
    Assumes that the list 'blocks' contains enough block TaskObjects to construct the specified
    structure. Assumes the blocks are homogeneous in size.
```

```
      Arranges the blocks to form a 3D structure of the given dimensions.
      """
      if len(dimensions) != 3:
          raise ValueError("Dimensions should be a tuple of three integers.")

      block_size = get_object_size(blocks[0])
      block_index = 0

      for z in range(dimensions[2]):
          layer_blocks = blocks[block_index : block_index + dimensions[0] * dimensions[1]]
          layer_start_position = get_point_at_distance_and_rotation_from_point(
              pose.position,
              pose.rotation,
              (block_size[2] + gap) * z,
              direction=np.array([0, 0, 1]),
          )
          layer_start_pose = Pose(layer_start_position, pose.rotation)
          for y in range(dimensions[1]):
              row_blocks = layer_blocks[y * dimensions[0] : (y + 1) * dimensions[0]]
              row_start_position = get_point_at_distance_and_rotation_from_point(
                  layer_start_pose.position,
                  layer_start_pose.rotation,
                  (block_size[1] + gap) * y,
                  direction=np.array([0, 1, 0]),
              )
              row_start_pose = Pose(row_start_position, layer_start_pose.rotation)
              make_line_with_blocks(row_blocks, row_start_pose, gap=gap)
          block_index += dimensions[0] * dimensions[1]
```

Listing 15: Function place_roof_tiles

```
def place_roof_tiles(roof_tiles: list[TaskObject], specific_pose: Pose):
    """
    Places exactly six roof tiles starting from a specific pose.
    Arranges the roof tiles evenly from the specified starting position and orientation.
    Ensures that the list of roof tiles has exactly six elements before proceeding.
    Args:
    - roof_tiles (list[TaskObject]): A list of TaskObjects identified as roof tiles.
    Must contain exactly six tiles.
    - specific_pose (Pose): The Pose representing the starting position
    and orientation for tile placement.
    """
    if len(roof_tiles) != 6:
        raise ValueError("There must be exactly six roof tiles")
    tile_width = roof_tiles[0].size[0]

    # Rotate the tiles by 90 degrees relative to the starting pose
    relative_rotation = specific_pose.rotation * Rotation.from_euler('z', 90, degrees=True)
    adjusted_pose_left = Pose(
        Point3D(specific_pose.position.x, specific_pose.position.y-tile_width/2, specific_pose.position.z),
        relative_rotation
    )
    put_first_on_second(get_object_pose(roof_tiles[0]), adjusted_pose_left)
    # Place one block on either side of the first block on the left
    move_block_next_to_reference(roof_tiles[1], roof_tiles[0], axis='-y', gap=0.005)
    move_block_next_to_reference(roof_tiles[2], roof_tiles[0], axis='y', gap=0.005)
    # Place the first block of the second set to the right (x direction) of the first block of the first set
    move_block_next_to_reference(roof_tiles[3], roof_tiles[0], axis='x', gap=0.005)
    # Place one block on either side of the second set
    move_block_next_to_reference(roof_tiles[4], roof_tiles[3], axis='-y', gap=0.005)
    move_block_next_to_reference(roof_tiles[5], roof_tiles[3], axis='y', gap=0.005)
```

Listing 16: Function make_line_of_blocks_next_to

```
def make_line_of_blocks_next_to(blocks: list[TaskObject], referenceBlock: TaskObject,
                                direction: str, gap: float = 0.005):
    """
    Arranges the given blocks in a straight line next to a reference block in the
    specified direction.
    Args:
        blocks (list[TaskObject]): A list of TaskObject instances representing
        the blocks to be arranged in a line.
        referenceBlock (TaskObject): The TaskObject representing the reference
        block next to which the line will start.
        direction (str): A string indicating the direction in which to align the line of blocks.
                        Valid directions are "front", "back", "left", and "right".
        gap (float): The gap between the reference block and the first block in the line,
        and between consecutive blocks.
    This function will arrange the specified blocks in a single line, starting from the chosen
    side of the reference block, following the given direction along the x or y axis in the workspace,
    depending on the specified direction.
    """
    axis = ''
    if direction == "front":
        axis = 'x'
    elif direction == "back":
        axis = '-x'
    elif direction == "left":
        axis = '-y'
```

```
1674
1675      elif direction == "right":
              axis = 'y'
1676      else:
              raise ValueError("Invalid direction provided. Use 'front', 'back', 'left', or 'right'.")
1677      current_reference = referenceBlock
          for block in blocks:
1678          move_block_next_to_reference(block, current_reference, axis=axis, gap=gap)
              current_reference = block
1679
```

Listing 17: Function make_line_with_blocks

```
1682      from utils.core_types import TaskObject, Pose
          from utils.core_primitives import (
1683          get_object_pose,
              get_object_size,
1684          put_first_on_second,
              get_point_at_distance_and_rotation_from_point,
1685      )

1686      def make_line_with_blocks(
1687          blocks: list[TaskObject], start_pose: Pose, gap: float = 0.005
          ):
1688          """Arranges the given blocks in a straight line starting from the specified start pose.
1689          Args:
                  blocks (list[TaskObject]): A list of block objects to be arranged in a line.
1690              start_pose (Pose): The pose in the workspace where the line of blocks should start.
                                      The position will be used as the starting point, and the rotation
1691                                  will be used as the direction vector.
              gap (float): The gap between consecutive blocks.
1692          Note:
              The function places the blocks in the order in which they are passed.
1693          """
              current_block = blocks[0]
1694          put_first_on_second(get_object_pose(current_block), start_pose)
              for block in blocks[1:]:
1695              # Get the current pose of the block
                  move_block_next_to_reference(block, current_block, axis="x", gap=gap)
1696              current_block = block
1697
```

Listing 18: Function stack_blocks

```
1700      def stack_blocks(blocks: list[TaskObject], start_pose: Pose):
1701          """Stacks a sequence of blocks on top of each other starting from a specified pose.
              Args:
1702          blocks: A list of TaskObject instances representing the blocks to be stacked.
              start_pose: A Pose indicating the initial position and orientation of the bottom block.
1703          """
              # Check if there's already a block in the starting position
1704          all_blocks = get_objects()
1705          for block in all_blocks:
                  block_pose = get_object_pose(block)
1706              # Check if block is close enough to the starting position, considering some tolerance
                  if (
1707                  abs(block_pose.position.x - start_pose.position.x) < 0.05 and
                      abs(block_pose.position.y - start_pose.position.y) < 0.05 and
1708                  abs(block_pose.position.z - start_pose.position.z) < 0.05
                  ):
1709                  # Move the block to the top-left corner (back_left)
                      workspace = Workspace()
1710                  top_left_position = workspace.back_left
                      put_first_on_second(block_pose, Pose(top_left_position, Rotation.identity()))
1711                  break
1712
              current_pose = start_pose
1713          for block in blocks:
                  pick_pose = get_object_pose(block)  # Get current pose of the block
1714              put_first_on_second(pick_pose, current_pose)  # Place block on the current pose
                  # Get the size of the block to calculate the new position for the next block
1715              block_size = get_object_size(block)
                  # Update the current pose to place the next block on top
1716              current_pose = Pose(
                      position=Point3D(
1717                      x=current_pose.position.x,
                          y=current_pose.position.y,
1718                      z=current_pose.position.z
                          + block_size[2],  # Increment z by block height
1719                  ),
                      rotation=current_pose.rotation,
1720              )
1721
```

Listing 19: Function move_block_next_to_reference

```
1726      def move_block_next_to_reference(
1727          block: TaskObject, referenceBlock: TaskObject, axis: str = "x", gap: float = 0.005
          ):
              """Moves the block next to the referenceBlock such that their edges are aligned along
              the specified axis with a small gap.
```

```
    Args:
        block (TaskObject): The block object to be moved and aligned.
        referenceBlock (TaskObject): The block object that remains stationary and serves as the reference.
        axis (str): The axis along which to align the blocks. Should be 'x', '-x', 'y', or '-y'.
        gap (float, optional): The small gap to leave between the blocks. Defaults to 0.005 meters.
    Raises:
        ValueError: If the specified axis is not 'x', '-x', 'y', or '-y'.
    """
    if axis not in ["x", "-x", "y", "-y"]:
        raise ValueError("Axis must be either 'x', '-x', 'y', or '-y'.")
    # Get the pose and size of the blocks
    block_pose = get_object_pose(block)
    reference_pose = get_object_pose(referenceBlock)
    reference_size = get_object_size(referenceBlock)
    block_size = get_object_size(block)
    # Determine the offset distance based on the axis
    if axis == "x":
        offset = (reference_size[0] + block_size[0]) / 2 + gap
        direction = np.array([1, 0, 0])   # positive x-axis direction
    elif axis == "-x":
        offset = (reference_size[0] + block_size[0]) / 2 + gap
        direction = np.array([-1, 0, 0])  # negative x-axis direction
    elif axis == "y":
        offset = (reference_size[1] + block_size[1]) / 2 + gap
        direction = np.array([0, 1, 0])   # positive y-axis direction
    elif axis == "-y":
        offset = (reference_size[1] + block_size[1]) / 2 + gap
        direction = np.array([0, -1, 0])  # negative y-axis direction

    rotated_direction = reference_pose.rotation.apply(direction)
    new_position = reference_pose.position.np_vec + offset * rotated_direction
    new_position = Point3D.from_xyz(new_position)
    # New pose for the block with the same rotation as the reference
    new_pose = Pose(position=new_position, rotation=reference_pose.rotation)
    # Move the block
    put_first_on_second(block_pose, new_pose)
```

Listing 20: Applied core primitives

```
def get_object_pose(obj: TaskObject) -> Pose:
    """returns the pose (Point3d, Rotation) of a given object in the environment."""
    return _from_pybullet_pose(env.get_object_pose(obj.id))

def put_first_on_second(pickPose: Pose, placePose: Pose):
    """
    This is the main pick-and-place primitive.
    It allows you to pick up the TaskObject at 'pickPose', and place it at the Pose specified by 'placePose'.
    If 'placePose' is occupied, it places the object on top of 'placePose.
    """
    return env.step(
        action={
            "pose0": _to_pybullet_pose(pickPose),
            "pose1": _to_pybullet_pose(placePose),
        }
    )

def get_object_size(task_object: TaskObject) -> tuple[float, float, float]:
    """Returns the size of the given TaskObject as a tuple (width, depth, height)."""
    return task_object.size

def get_object_color(task_object: TaskObject) -> str:
    """Returns the color of the given TaskObject."""
    return task_object.color

def get_objects() -> list[TaskObject]:
    """gets all objects in the environment"""
    return env.task.taskObjects
```

## F    REAL-WORLD DEMONSTRATIONS

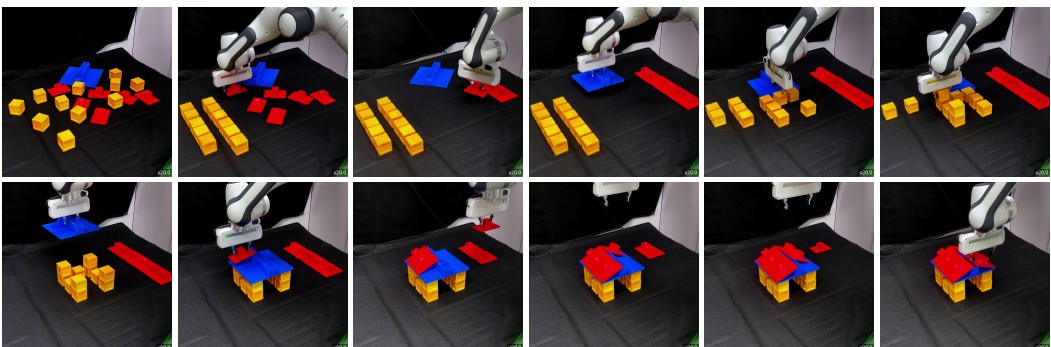

Figure S11: Real-world demonstration for task "build a house".

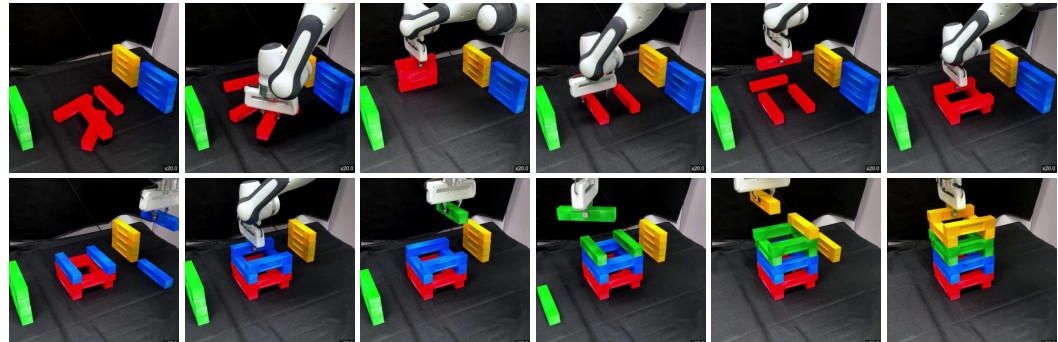

Figure S12: Real-world demonstration for task "stack a jenga tower".

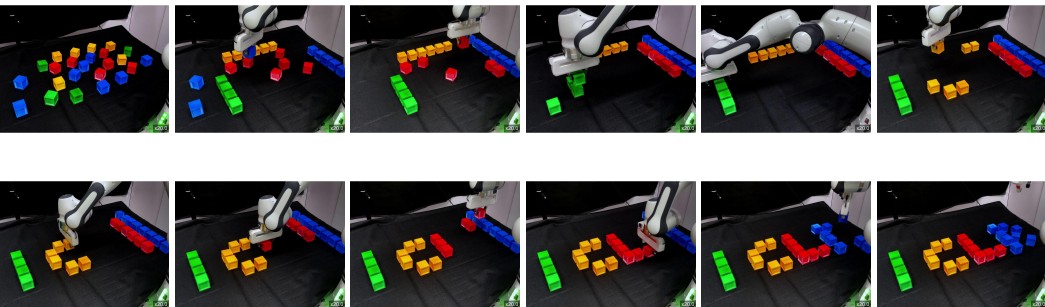

Figure S13: Real-world demonstration for task "write a ICLR".

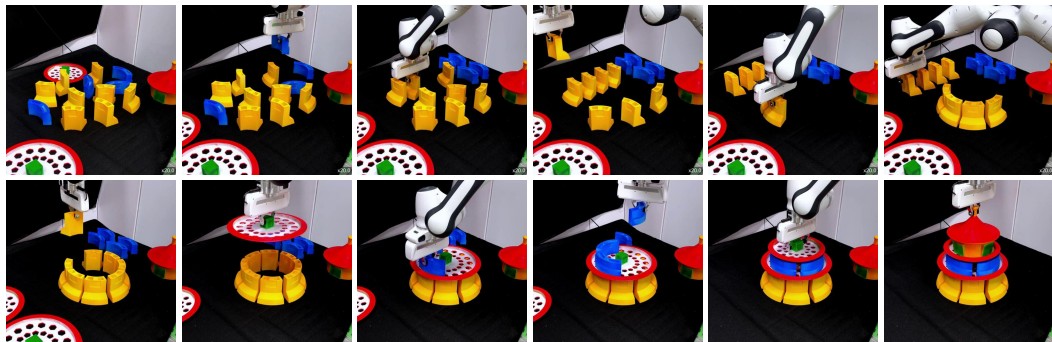

Figure S14: Real-world demonstration for task "build a temple".

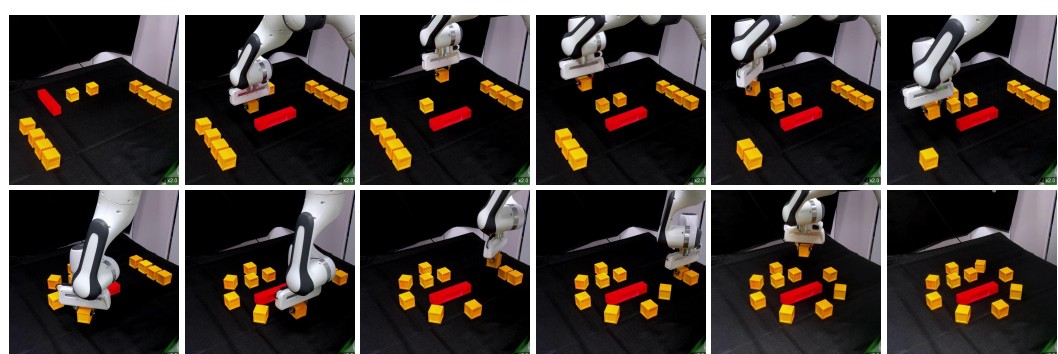

Figure S15: Real-world demonstration for task "build a human face".

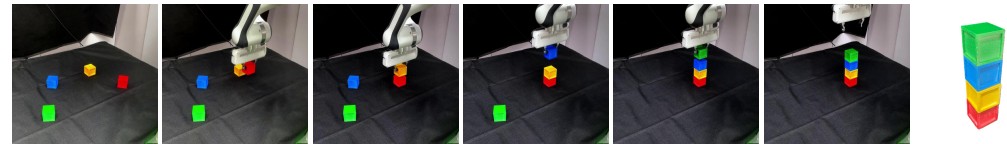

Figure S16: Real-world demonstration for task "stack blocks" with corner-to-corner aligned.

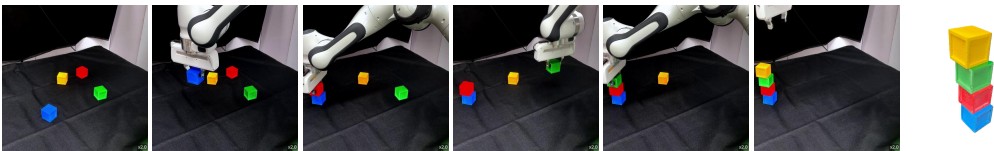

Figure S17: Real-world demonstration for task "stack blocks" with shift 0.5cm from each other.

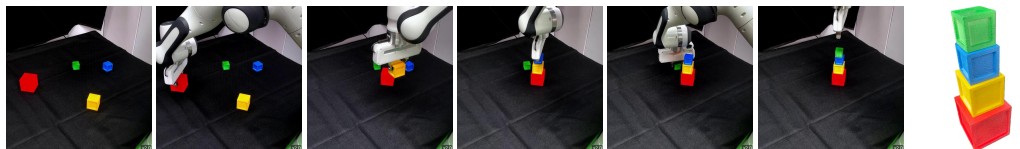

Figure S18: Real-world demonstration for task "stack blocks" from big to small.

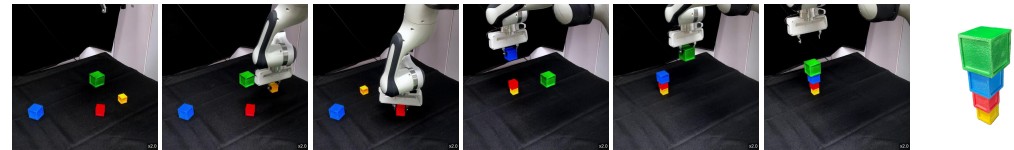

Figure S19: Real-world demonstration for task "stack blocks" from small to big.

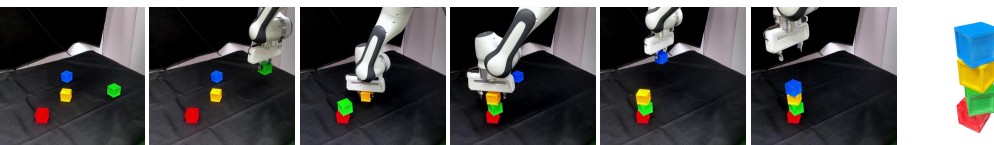

Figure S20: Real-world demonstration of the "stack blocks" task, showcasing the performance of the baseline models, e.g., CaP or DAHLIA.

# G  ADDITIONAL EMPIRICAL ANALYSIS

## BASELINE COMPARISON

Table S4 presents the empirical comparison between three representative hierarchical and lifelong RL baselines and our proposed framework LYRA across 10 Meta-World manipulation tasks. For RL baselines, each method is trained for 1 million steps per task, and evaluated using five random seeds, reporting average success rates. In our experiment, we include the following baselines:

- **HiLL** Zhang et al. (2023a): A hierarchical RL framework that leverages a latent landmark graph to balance RL exploration and exploitation.
- **Text2Reward** Xie et al. (2024): An LLM-guided reward refinement framework that generates dense reward trajectories for RL.
- **LEGION** Meng et al. (2025a): A state-of-the-art lifelong RL framework that leverages a Bayesian Non-parametric model for dynamic knowledge preservation and adaptation.

For HiLL and Text2Reward, which are designed for standard single-task RL, we report their fine-tuned performance on each task independently. For the lifelong RL baseline LEGION, we report the best performance achieved across its asynchronous learning process. We additionally evaluate "Forgetting" as defined in Meng et al. (2025a), where the scalar ranges from 0 to 1 (0 = no forgetting, 1 = complete forgetting). For our framework LYRA, we report performance under two settings: $\leq 3$ number of corrections (NoC) and $\leq 5$ NoC.

The RL baselines exhibit mixed performance across individual tasks, with average success rates of 0.64 (HiLL), 0.78 (Text2Reward), and 0.72 (LEGION) after 1M training steps per task. In contrast, LYRA achieves an average success rate of 0.80 using no more than three human corrections, and reaches a 1.0 success rate on all tasks with at most five corrections. For relatively simple primitives such as Reach, Door-open, and Button-press, LYRA achieves perfect success even without any human intervention, indicating that the learned skill library generalizes robustly to these scenarios. We also evaluate knowledge preservation using the Forgetting metric defined in LEGION, where lower values indicate better retention. LYRA achieves a perfect score of 0.00, comparable to LEGION's 0.02, demonstrating that its externally stored skill representations avoid the catastrophic forgetting common in lifelong RL. While the RL methods perform reasonably well on short-horizon, single-primitive tasks, they remain fundamentally limited in long-horizon reasoning and fail entirely on multi-stage Ravens tasks such as Build a House. These results highlight the efficiency and robustness of LYRA's correction-driven, skill-based paradigm and underscore the difficulty current RL systems face in scaling to extreme long-horizon manipulation.

Table S4: Comparison of hierarchical and lifelong RL baselines and LYRA on Meta-World tasks, including success rates and forgetting.

| Method | Success rate(↑) | | | | | | | | | | | Forgetting(↓) |
|---|---|---|---|---|---|---|---|---|---|---|---|---|
| | Reach | Push | Sweep-into | Door-open | Faucet-open | Drawer-close | Button-press | Coffee-pull | Window-close | Hammer | Total average | |
| HiLL | 1.0 | 0.6 | 0.4 | 0.8 | 0.8 | 0.8 | 1.0 | 0.2 | 0.8 | 0.0 | 0.64 | N/A |
| Text2Reward | 1.0 | 0.6 | 0.6 | 1.0 | 1.0 | 0.8 | 1.0 | 0.6 | 0.8 | 0.4 | 0.78 | N/A |
| LEGION | 0.8 | 1.0 | 0.8 | 0.6 | 1.0 | 0.6 | 0.8 | 0.4 | 1.0 | 0.2 | 0.72 | 0.02 |
| LYRA ($\leq 3$ NoC) | 1.0 | 0.0 | 1.0 | 1.0 | 1.0 | 1.0 | 1.0 | 1.0 | 1.0 | 0.0 | 0.80 | 0.00 |
| LYRA ($\leq 5$ NoC) | 1.0 | 1.0 | 1.0 | 1.0 | 1.0 | 1.0 | 1.0 | 1.0 | 1.0 | 1.0 | 1.00 | 0.00 |

Table S5 reports the comparison between two recent large-scale VLA models (OpenVLA and $\pi_0$) and our proposed framework across short- and long-horizon manipulation benchmarks. Specifically, we evaluated all methods on five Meta-World (MW) tasks to validate short-horizon performance, and further compare them on LIBERO-long as well as our customized extreme long-horizon Ravens tasks (e.g., building a jenga tower, constructing a 4×4×3 pyramid, and building a house). For consistency, in Metaworld and our Ravens tasks, we collected 30 trajectories per task using our skill-based code generation framework, recording synchronized third-person views and wrist-camera frames (224×224 px). The collection and data preprocessing costs are approximately 5 hours. We used joint-space actions for all VLA models. Both OpenVLA and pi0 were fine-tuned for 50k steps using LoRA on their officially released LIBERO-pretrained weights, following the original hyperparameters. Training requires 4 × A100 GPUs (40 GB), totaling approximately 50–60 GPU hours.

As shown in the table, on the five short-horizon Meta-World tasks, both OpenVLA and $\pi_0$ achieve performance comparable to LYRA, indicating that pretrained VLA policies are reliable in simple single-primitive settings. However, their performance degrades substantially when evaluated on long-horizon tasks. Even on LIBERO-long, where each task requires only two sequential primitives, OpenVLA and $\pi_0$ reach success rates of only 0.65 and 0.60, respectively. On our customized Ravens benchmarks, which require over 20 primitives and significantly richer spatial reasoning, both models fail entirely, achieving 0.0 success on tasks such as build a jenga tower, construct a 4×4×3 pyramid, and build a house. In contrast, LYRA successfully completes all tasks using its library of compositional and reusable skills, reaching success rates between 0.80 and 0.95 across these extreme long-horizon scenarios for task plan generation. We attribute the VLA failures to two factors: (1) data sparsity, as VLA models require hundreds of expert demonstrations even for two-primitive tasks and still exhibit limited generalization; and (2) restricted distributional diversity in existing VLA training benchmarks, which often assume narrow object initialization ranges. By comparison, our Ravens environments sample object placements across the full workspace, making visual and spatial generalization more demanding. These findings highlight the limitations of current large-scale VLA models in extreme long-horizon settings and demonstrate the advantages of LYRA's skill-based, correction-efficient approach for robust compositional manipulation.

Table S5: Comparison between OpenVLA, $\pi_0$, and LYRA on Meta-World, LIBERO-long, and long-horizon Ravens tasks.

| | MW – Pick-place | MW – Door-open | MW – Window-close | MW – Coffee-pull | MW – Hammer | LIBERO – long | Build a Jenga tower | Construct a 4×4×3 pyramid | Build a House |
|---|---|---|---|---|---|---|---|---|---|
| OpenVLA | 0.95 | 0.90 | 1.00 | 0.95 | 0.85 | 0.65 | 0.00 | 0.00 | 0.00 |
| $\pi_0$ | 0.90 | 1.00 | 1.00 | 0.90 | 0.90 | 0.60 | 0.00 | 0.00 | 0.00 |
| LYRA (Ours) | 1.00 | 1.00 | 1.00 | 1.00 | 1.00 | 0.95 | 0.95 | 0.85 | 0.80 |

### TIME EFFICIENCY COMPARISON

To quantify the time cost associated with human-in-the-loop feedback, we analyze timestamped logs from 30 feedback loops collected in the Ravens benchmark, data collected from the same user. Our interactive terminal UI allows users to input free-form text in real time, enabling precise measurement of the full human feedback cycle. Because the interface provides predefined interaction options, we use three timestamps as reference points: (1) $t_1$, the moment the task execution finishes; (2) $t_2$, the moment the user selects the "Provide Feedback" option; and (3) $t_3$, the moment the feedback is submitted. The interval $t_2 - t_1$ represents the user's reasoning and decision-making time, and $t_3 - t_2$ represents the time required to type and submit the feedback. Table S6 reports the measured approximate time cost across the full feedback loop, including the decision-making phase, and the text-entry or text-generation phase. A complete human feedback cycle, including both reasoning and typing, takes on average 30.5 seconds, while an LLM produces feedback in 15.8 seconds, with overlapping minimum and maximum ranges. When isolating the actual decision-making time, humans are consistently faster, requiring less than 8 seconds to determine the correct correction, whereas LLMs require more than 14 seconds for inference and reasoning. The additional time on the human side stems primarily from keyboard typing (over 75% of total time), which reflects typing speed rather than cognitive load. These results indicate that the human cognitive burden during feedback is low, and the time difference between human and LLM feedback is incremental rather than fundamental. Although LLMs generate text faster, their feedback must adhere to structured formats in prior works Meng et al. (2025b); Zhang et al. (2023b); Zhi et al. (2025), limiting expressiveness and generalization. In contrast, our use of free-form human feedback offers far richer and more adaptive guidance, which we find essential for enabling the agent to escape local optima and acquire robust, reusable skills in long-horizon settings.

Table S6: Human-in-the-loop vs. LLM response time analysis.

| | Human user | LLMs |
|---|---|---|
| Average time of a feedback loop | 30.5 sec (Min 17.2 sec, Max 39.5 sec) | 15.8 sec (Min 4 sec, Max 32.4 sec) |
| Average time for reasoning/decision | 8 sec (below 25%) | 14 sec (over 90%) |
| Average time for text typing/generation | 22 sec (over 75%) | 2 sec (below 10%) |

