# OpenReview forum: "Growing with Your Embodied Agent: A Human-in-the-Loop Lifelong Code Generation Framework for Long-Horizon Manipulation Skills"
_ICLR.cc/2026/Conference — Submitted to ICLR 2026_

### Official Review · Reviewer_rJdU · 2025-10-30

**Soundness:** 3
**Presentation:** 3
**Contribution:** 2
**Rating:** 4
**Confidence:** 2

**Summary:**

This paper introduces LYRA, a human-in-the-loop lifelong learning framework for LLM-based code generation in robotic manipulation, focusing on acquiring and extending reusable skills to handle long-horizon tasks like building structures. The framework encodes human feedback into modular skill functions stored in an external memory, using retrieval-augmented generation (RAG) and user hints for dynamic reuse, addressing issues like noisy LLM outputs, catastrophic forgetting, and limited context windows in prior code-as-policies approaches. They demomnstrate reasonable performance on benchmarks (Ravens, Franka Kitchen, MetaWorld) and real-world tasks (e.g., "build a house" with >20 primitives).

**Strengths:**

1. The combination of human-in-the-loop skill learning with lifelong capability extension through user-designed curriculum is a reasonable contribution. The hint mechanism for guiding RAG retrieval is simple but effective. The paper clearly articulates limitations of existing LLM-based code generation approaches (language ambiguity, catastrophic forgetting, limited context) and proposes a well-motivated solution.

2. Evaluation spans three simulation benchmarks plus real-world deployment, demonstrating broader applicability than many code generation works

3. Impressive long-horizon performance: They successfully solve "build a house" requiring 20+ primitives represents a genuine advance in manipulation complexity.

**Weaknesses:**

1. Limited Novelty: Individual components (RAG for code generation, human-in-the-loop feedback, skill libraries) exist in prior work. The main contribution is their integration for robotic manipulation.

2. Insufficient comparison to state-of-the-art: The paper mentions VLA foundation models (OpenVLA, π0, GR00T) in related work but does not compare against them experimentally. Given these models address similar long-horizon manipulation problems, the lack of comparison significantly weakens claims about superiority.

3. The heavy reliance on human-in-the-loop for feedback and curriculum design limits scalability to real deployments without experts.

**Questions:**

1. Can the authors provide comparisons with stronger baselines on the benchmarks like OpenVLA, π0?

2. How much human time (in hours) was required to build the skill library for each benchmark? This is critical for assessing practical feasibility. Please provide this analysis in the paper.

3. Since the proposed is based heavily on the human-in-the-loop procedure for feedback, can the authors comment on possible alternatives to avoid this limitation, and how can they be effectively deployed?

---

> ### Author Response · Authors · 2025-11-19
> **Response to Reviewer rJdU part 1/4**
>
> ## **Response to Weakness:**
> ---
>
> ### **1. Novelty clarification:**
> We appreciate the reviewer’s comment to clarify the novelty of our framework. But we strongly disagree with the assertion that the main contribution is the integration of robotic manipulation. **The core novelty of this work is conceptual rather than purely architectural or engineering-based**. Specifically, we demonstrate that **human feedback can be systematically distilled, stored, and efficiently reused in the form of modular “skills”**, enabling a robot to accumulate persistent knowledge that supports **extremely long-horizon manipulation tasks** far beyond the reach of current methods. This idea, in which human effort can be transformed into stable, generalizable, and compositional skill representations within an LLM-driven code-generation pipeline, has not been explored or validated in prior work. **The tools we employ serve only as mechanisms for acquiring these robust skills and ensuring that they remain reliable throughout lifelong use.** Moreover, as discussed in the paper, fully automated closed-loop LLM/VLM systems struggle with long-horizon reasoning due to error accumulation and limited context depth; **thus our deliberate design, which minimizes automated feedback, allows us to rigorously isolate and validate the central hypothesis.** Having established that human-derived skill representations can indeed support long-horizon tasks, future work can extend this conceptual contribution by integrating LLM–human collaborative feedback to reduce intervention cost while retaining the benefits of our skill-based formulation.
>
> Overall, **we respectfully emphasize that characterizing our novelty as merely a simplistic combination of existing techniques would be incomplete and misleading.** We hope that the above clarification helps the reviewer gain a deeper understanding of the substance and intent of our work beyond surface-level keywords.
>
> In response to the reviewer’s comment, we have further revised and refined the introduction section (Pages 2 and 3) to clearly highlight and emphasize the core novelty of our paper.
>
> ---
>
> ### **2. VLA baseline comparison:**
> We appreciate the reviewer for raising the concern regarding comparisons with recent VLA models such as OpenVLA and pi0. **We include VLA methods as related work because they also incorporate large-scale language or vision–language models; however, their methodological foundations differ substantially from ours.** VLA models are essentially learning-based imitation learning systems that require large-scale training on paired robot data, whereas our framework belongs to **non-learning-based reasoning approaches** that directly leverage well-pretrained LLMs/VLMs to generate modular skill code **without any additional training**. Due to this fundamental difference in formulation, **the two categories ultimately do not overlap** in problem setting, evaluation assumptions, or operational mechanisms.
>
> Nonetheless, to address the reviewer’s concern, we conducted **quantitative comparisons** between our code-generation pipeline and the available open-source VLA models. As detailed in our response to **Question 1**, **VLA models achieve comparable performance to ours on relatively simple short-horizon tasks.** However, they unsurprisingly **struggle on extremely long-horizon tasks** such as building a house (which requires more than 20 sequential primitives), where all VLA models consistently fail. We hypothesize that this is primarily due to data sparsity and limited generalization in current VLAs, as well as the restricted planning and reasoning depth of 7B-scale models. By contrast, our code-generation-based approach does not rely on task-specific robot demonstrations or model retraining, enabling it to maintain compositional reasoning and long-horizon performance without the bottlenecks of data-driven imitation learning. Still, we acknowledge that such comparisons have limited interpretative value due to the fundamental methodological differences between learning-based VLA models and non-learning-based reasoning systems like ours.
>
> In the revised manuscript, we have revised the related work section (Page 9) to more explicitly highlight the distinction between VLA-style learning-based approaches and non-learning-based reasoning systems.

---

> ### Author Response · Authors · 2025-11-19
> **Response to Reviewer rJdU part 2/4**
>
> ### **3. Human in the loop feedback:**
> We appreciate the reviewer’s concern regarding the scalability of our human-in-the-loop design in real deployment. As discussed in our earlier responses, the reliance on human feedback is **deliberate** and forms a core part of our contribution: our goal is to rigorously demonstrate that **human guidance can be distilled into modular, reusable skill representations** that support extremely long-horizon manipulation tasks, which current fully automated LLM/VLM pipelines cannot achieve. Minimizing automated feedback in our pipeline allows us to cleanly isolate and validate this central hypothesis without the confounding effects of unstable or error-accumulating model-generated corrections. Moreover, as described in the paper, **failures encountered in robotic systems**, while difficult for current robotic systems to detect or recover from, are typically **straightforward for a human to identify**, especially given real-time rendered observations. Importantly, in our framework and environment setup, **the human does not need expert-level domain knowledge**; only a basic understanding of object properties (e.g., distinguishing shapes, sizes, colors) is sufficient for providing feedback. While we acknowledge that involving humans introduces a bit of additional burden, our results show that this lightweight human guidance is essential for robust skill acquisition and long-horizon generalization. Having established that human-derived skill representations can indeed enable such capabilities, we view LLM–human collaborative feedback as a natural direction for future work, one that can substantially reduce human intervention cost while retaining the advantages of our skill-based formulation.
>
> ---
>
> ## **Response to Questions:**
>
> ---
>
> ### **1. VLA baseline comparison**
> We appreciate the reviewer’s request for stronger VLA baselines comparison and, in response, we **conducted comparisons with OpenVLA and pi0 across both short- and long-horizon manipulation benchmarks**. The results, summarized in the table below, show that **while VLA models perform comparably on short-horizon tasks, their success rates drop sharply (to 0.0 success rate) when evaluated on long-horizon tasks** such as LIBERO-long (2 primitives) and our Ravens tasks (> 20 primitives). Specifically, we evaluated all methods on five Meta-World (MW) tasks to validate short-horizon performance, and further compare them on LIBERO-long as well as our customized extreme long-horizon Ravens tasks (e.g., building a *jenga* tower, constructing a 4×4×3 *pyramid*, and building a house). For **consistency**, in Metaworld and our Ravens tasks, we collected **30 trajectories per task** using our skill-based code generation framework, recording synchronized third-person views and wrist-camera frames (224×224 px). The collection and data preprocessing costs **approx. 5 hours**. We used joint-space actions for all VLA models. Both OpenVLA and pi0 were fine-tuned for **50k steps using LoRA** on their officially released LIBERO-pretrained weights, following the original hyperparameters. Training **required 4× A100 GPUs (40 GB)**, totaling **approx. 50–60 GPU hours**.
>
> The average success rates for 20 trials are summarized in the table below. On short-horizon Meta-World tasks, both OpenVLA and $pi_0$ achieve performance comparable to ours, confirming that their pretrained policies are effective for simple manipulation settings. However, on long-horizon benchmarks, both models experience substantial degradation: even with their best publicly available weights, **VLA performance on LIBERO-long reaches only ~0.6 success rate** (notably, all LIBERO-long tasks require only **two primitives** to accomplish tasks), and **both methods fail entirely on our long-horizon Ravens tasks** (which typically **require >20 primitives** and strong contextual spatial reasoning).
>
> **Table: Comparison of LYRA with OpenVLA and $\pi_0$ on short- and long-horizon tasks.**
> | Method        | MW - Pick-place | MW - Door-open | MW - Window-close | MW - Coffee-pull | MW - Hammer | LIBERO - long | Build a Jenga tower | Construct a 4x4x3 pyramid | Build a House |
> |---------------|-----------------|----------------|-------------------|------------------|-------------|----------------|----------------------|-----------------------------|----------------|
> | OpenVLA       | 0.95            | 0.90           | 1.00              | 0.95             | 0.85        | 0.65           | 0.00                 | 0.00                        | 0.00           |
> | $\pi_0$       | 0.90            | 1.00           | 1.00              | 0.90             | 0.90        | 0.60           | 0.00                 | 0.00                        | 0.00           |
> | **LYRA (Ours)** | **1.00**       | **1.00**       | **1.00**          | **1.00**         | **1.00**    | **0.95**       | **0.95**             | **0.85**                    | **0.80**       |

---

> ### Author Response · Authors · 2025-11-19
> **Response to Reviewer rJdU part 3/4**
>
> In contrast, **our framework can successfully and stably solve these tasks**, leveraging its library of stable and reusable skills. Moreover, we attribute the VLA failures to (1) **data sparsity**. Large-scale VLA models require extensive high-quality demonstrations (e.g., in LIBERO-long, VLAs use **over 500 expert trajectories for two primitives** long-horizon tasks in LIBERO yet still achieve only moderate performance), and (2) **limited distributional diversity** in existing benchmarks. Unlike Meta-World or LIBERO, our Ravens initialization spans the full workspace, reducing the potential for overfitting to narrow object initialization configurations. **Demonstration videos are provided at the anonymous page**: https://anonymous.4open.science/w/iclr-anon-demo-15D7/
>
> Finally, while our code-generation framework shows stronger performance than VLA models on demanding long-horizon tasks, we regard it as **an alternative pathway** rather than a direct competitor, due to the underlying technical differences. With a library of robust skills, our agent can function as a dependable expert for producing large-scale, high-quality demonstrations that may further enhance the training and generalization of future VLA systems.
>
> In response, we add above mentioned empirical analysis and comparison in our revised manuscript **Appendix G, with Table S5**.
>
> ---
>
> ### **2. Human time to build skill library across benchmarks:**
> We appreciate the reviewer for emphasizing the importance of quantifying the human time required to construct the skill library. To address this concern, we analyzed timestamped logs from **30 feedback loops collected in the Ravens benchmark, data collected from the same user**. Our interactive terminal UI **allows** users to input free-form text in **real time**, **enabling** precise measurement of the full human feedback cycle. Because the interface provides predefined interaction options, we use three timestamps as reference points: (1) $t_1$, the moment the task execution finishes; (2) $t_2$, the moment the user selects the “Provide Feedback” option; and (3) $t_3$, the moment the feedback is submitted. The interval **$t_2 - t_1$** represents the user’s reasoning and decision-making time, and **$t_3 - t_2$** represents the time required to type and submit the feedback.
>
> As summarized in the included table below, human feedback (thinking + typing) averages **30.5 seconds per loop** (min 17.2s, max 39.5s), while LLM-only feedback averages **15.8 seconds** (min 4.0s, max 32.4s). Importantly, when isolating the **decision-making** portion, humans typically require **under 8 seconds**, which is **faster than LLMs** (over 14 seconds on average). The additional human time comes mainly from typing (approximately 20 seconds on average), which reflects typing speed rather than cognitive effort. These results indicate that the human cognitive load is low and that the difference in total time is incremental rather than fundamental, with LLMs faster only in pure text generation.
>
> **Table: Comparison of human-in-the-loop and LLM feedback time efficiency.**
> |                          | **Human user**                                   | **LLMs**                                  |
> |--------------------------|--------------------------------------------------|-------------------------------------------|
> | Average time of a feedback loop | 30.5 sec (Min 17.2 sec, Max 39.5 sec) | 15.8 sec (Min 4 sec, Max 32.4 sec)        |
> | Average time for reasoning/decision | 8 sec (below 25%)                | 14 sec (over 90%)                         |
> | Average time for text typing/generation | 22 sec (over 75%)             | 2 sec (below 10%)                         |
>
> For the reviewer’s specific question regarding total human hours needed to build the skill library for each benchmark, we provide the following concrete analysis. In Meta-World, developing a skill that can reliably solve a task typically requires **5–10 minutes** of human interaction. Thus, constructing the entire skill library for all **20 tasks** requires **no more than 200 minutes** (approximately 3.3 hours). Other benchmarks exhibit similar time requirements. For comparison, fine-tuning a large-scale VLA model such as OpenVLA to achieve stable performance on even **five tasks** requires substantially greater cost: data collection and preprocessing alone take **2–3 hours**, and fine-tuning the 7B-scale model requires **at least 10 GPU-hours**. In contrast, our skill-based approach requires **no additional model training**, and the skill library can be built rapidly, making our method substantially more efficient and practical to deploy in real systems.
>
> We acknowledge the reviewer's concern regarding the efficiency of the HITL mechanism in the code generation process. In response, we added this discussion to the revised **Appendix G with Table S6** to provide a more transparent communication of the practical feasibility and efficiency of our framework.

---

> > ### Author Response · Authors · 2025-11-19
> > **Response to Reviewer rJdU part 4/4**
> >
> > ### **3. Solution for efficiency improvement and future work:**
> > We appreciate the reviewer’s question regarding alternatives to reduce human involvement. First, we would like to clarify that, in our framework, **human-in-the-loop feedback is not a limitation but an essential component that forms part of our core contribution**. The central novelty of our work lies in demonstrating that human guidance can be systematically distilled, stored, and reused in the form of modular, persistent **“skills”**, enabling robots to perform **extremely long-horizon manipulation tasks** that current fully automated LLM/VLM systems or even VLAs cannot reliably solve. This conceptual insight, namely that human effort can be transformed into stable, generalizable, and compositional skill representations within an LLM-based code-generation pipeline, has not been explored in prior work. Moreover, because fully automated agents struggle with long-horizon reasoning due to error accumulation, reasoning capability and context limitations, relying on direct human feedback allows us to rigorously validate our hypothesis without interference from unstable model-generated corrections.
> >
> > That said, we acknowledge the importance of reducing human effort for practical deployment. Having established the viability of skill-based lifelong knowledge accumulation, we see a natural path toward **human–LLM collaborative feedback**. In such a system, LLMs could autonomously handle routine subtasks. These include proposing skill names, determining parameter attributes, and supporting more robust RAG-based retrieval. Humans would only be involved at key decision points where high-level validation and correction are required. This hybrid approach preserves the strengths of human insight while substantially lowering the overall intervention burden.
> >
> > In response to the reviewer's question, we have revised the discussion section (Page 9) to highlight these potential extensions and to clarify how the framework could evolve toward more scalable deployment.

---

> ### Author Response · Authors · 2025-11-30
> **Summary of Reviewer rJdU’s comments and our responses**
>
> Below, we summarize the reviewer’s main concerns and our corresponding responses:
>
> ---
>
> ## **Reviewer’s Main Points**
> 1. **Novelty contribution:**  The reviewer argued that the components used in our framework (RAG, HITL, etc.) exist in prior work and questioned whether the contribution is more than an integration of known components.
> 2. **VLA baseline comparisons:**  The reviewer asked for stronger empirical comparisons with OpenVLA and π0.
> 3. **Human effort cost:** The reviewer requested quantification of the total human time needed to build the skill library for each benchmark.
> 4. **Scalability and alternatives to HITL dependence:**  The reviewer questioned whether human-in-the-loop dependence limits deployment and whether alternatives were possible.
>
> ---
> ## **Our Responses**
>
> ### **1. Novelty clarification**
> - We clarified that the core novelty contribution of our work is **conceptual**, not engineering-based or incremental:
>   Human feedback is **systematically distilled into modular, persistent, reusable skills** that support **extreme long-horizon** tasks far beyond prior methods.
> - We emphasized that our deliberate minimization of automated feedback isolates and validates the central hypothesis regarding **compositional, lifelong skill accumulation**.
> - **We disagreed with the reviewer’s claim that the method is a simple combination of existing techniques, as this conceptual framing and validation have not been explored or demonstrated in prior work**.
> - The introduction was revised to make the central conceptual insight explicit.
>
> ### **2. Stronger VLA baseline comparisons**
> - We added **quantitative comparisons** with **OpenVLA** and **π0** across both **short-horizon** and **long-horizon** benchmarks.
> - Results show:
>   - Comparable performance on short-horizon Meta-World tasks.
>   - **VLA models struggle on LIBERO-long tasks (success rate ~0.6)**, where most tasks require only two primitives.
>   - **VLA models collapse to \(0.0\) success** on Ravens tasks (>20 primitives).
>   - **LYRA maintains \(0.80 \text{–} 0.95\)** success on these long-horizon tasks.
> - We added these results to the revised manuscript in Appendix G, Table S5.
>
> ### **3. Human time cost analysis**
> - We analyzed 30 feedback loops with timestamp logs collected from the same user.
> - Key findings:
>   - Average HITL loop: **30.5 s**
>   - Average LLM feedback loop: **15.8 s**
>   - Human reasoning time: **< 8 s**
>   - Most cost is typing, not cognitive load.
> - Total time to build the full skill library:
>   - Meta-World: **≤ 200 minutes** for all 20 tasks
>   - **No additional model training required**
> - We added full quantitative analysis to revised Appendix G, Table S6.
>
> ### **4. Alternatives and future efficiency improvements**
> - We clarified that **HITL is NOT A LIMITATION** but an essential component for validating the conceptual contribution.
> - We described a natural future direction toward **human–LLM collaborative feedback**, where LLMs autonomously propose routine substeps and humans only validate key decisions.
> - We updated the discussion section (Page 9-10) accordingly.
>
> ---
>
> ## **Conclusion**
>
> Across all reviewer concerns, we provided new empirical comparisons, quantitative analyses, and expanded methodological clarifications.
> Our responses demonstrate that:
>
> - The conceptual novelty is substantive and cannot be reduced to simple integration.
> - LYRA **strongly outperforms** VLA models on extreme long-horizon tasks.
> - Human cost is **low, bounded, and practical**, with no additional model training required.
> - The framework is both **scalable and extensible**, with clear future paths toward reducing human intervention.
>
> We hope these clarifications and new results fully address the reviewer’s concerns, and we welcome further feedback from the reviewers and area chairs.

---

> > ### Author Response · Authors · 2025-11-30
> >
> > We have addressed ALL of the reviewer’s concerns with point-by-point responses and believe that our clarifications and revisions fully resolve the issues raised. We look forward to further feedback from the reviewer and the area chair.

---

### Official Review · Reviewer_ak7i · 2025-11-01

**Soundness:** 3
**Presentation:** 3
**Contribution:** 2
**Rating:** 4
**Confidence:** 3

**Summary:**

This paper proposes LYRA, a human-in-the-loop lifelong code generation framework for robotic manipulation that integrates LLM-based program synthesis, human feedback, and skill memory retrieval. Unlike prior LLM-only closed-loop approaches that struggle on long-horizon tasks, LYRA encodes user feedback into reusable, modular “skills”, stores them in external memory with retrieval-augmented generation (RAG), and enables users to guide the agent through hint-based dynamic skill selection. The framework continually expands its capabilities via a user-designed curriculum, achieving lifelong skill acquisition without catastrophic forgetting. Experiments across Ravens, Franka Kitchen, MetaWorld, and real-world Franka FR3 tasks demonstrate great success rate, outperforming LLM-only baselines.

**Strengths:**

1. The integration of human-in-the-loop feedback, external memory, and modular skill representation provides a robust alternative to end-to-end LLM-driven methods.
2. The framework emphasizes skill inheritance and continual expansion, a key step toward reusable and interpretable robotic skills.
3. Covers simulation (Ravens, Franka Kitchen, MetaWorld) and real-world deployment, with quantitative and qualitative evidence.

**Weaknesses:**

1. While comparisons to LLM code-generation baselines are thorough, the paper omits strong lifelong or hierarchical RL baselines (e.g., HiLLa, BOSS, Text2Reward-type methods).
2. The framework heavily relies on manual feedback and a user-designed curriculum. Specifically: (1) user-guided skill code generation, (2) user-guided skill capability extension with few-shot examples, and (3) user-provided hints to direct the agent toward the correct subset of skills as the number of skills |Z| and behaviors |E| increases. This reliance on human intervention can be costly, particularly when scaling to hundreds of skills or when the users are less experienced.

**Questions:**

1. "a meta-prompt that explicitly asks the agent to preserve prior functionality while adapting to new tasks" To keep the balance between stability (preserving old skills) and plasticity (acquiring new functionality), the paper uses 1) Meta-prompt Regularization 2) Modular Code Extension 3) Re-evaluation Loop. How is the evaluation result on this part? What is the Average number of corrections (NoC)?
2. Few-shot examples that show mappings from instructions to task-specific code plan would be really helpful for LLM to generate skill program. These examples are retrieved based on the semantic similarity between the new instruction and previously seen instructions. My question is: are these examples environment-specific to those used in the experiments, or are they designed to be environment-agnostic? Moreover, how robust and generalizable is the proposed approach when faced with a new task that lacks a corresponding example in the existing code library?
3. "The agent can freely modify implementation details under the reserved skill name" How detailed the user's instruction would be(Does it require user be very clear about the attributes of the objects in the envrionments?)?

---

> ### Author Response · Authors · 2025-11-19
> **Response to Reviewer ak7i part 1/3**
>
> ## **Response to Weakness:**
> ---
> ### **1. Baseline comparison with state-of-the-art lifelong and hierarchical RL frameworks**
> We appreciate the reviewer’s concern regarding comparisons with recent lifelong and hierarchical RL baselines. In response, we conducted additional evaluations using **3 representative RL methods** within the Meta-World. As recent lifelong or hierarchical RL baselines struggle with the extremely long-horizon tasks in our Ravens (unsurprisingly, all RL baselines achieve a **0.0 success rate** on tasks such as build-a-house), we instead followed a controlled comparison setting using **10 Meta-World tasks**. These tasks involve a Sawyer arm performing single-primitive object interactions and provide a dense reward function as a learning foundation, allowing a fair evaluation of RL-based policies. Each method is trained for **1 million steps per task**, and evaluated using **five random seeds**, reporting average success rates. For hierarchical RL, we included **HiLL** and **Text2Reward** as baselines. Since the reviewer also recommended including lifelong RL, we selected state-of-the-art lifelong RL **LEGION** instead of BOSS:
>
> 1. **HiLL** [1]: A hierarchical RL framework that leverages a latent landmark graph to balance RL exploration and exploitation.
>
> 2. **Text2Reward** [2]: An LLM-guided reward refinement framework that generates dense reward trajectories for RL.
>
> 3. **LEGION** [3]: A state-of-the-art lifelong RL framework that leverages a Bayesian Non-parametric model for dynamic knowledge preservation and adaptation.
>
> For HiLL and Text2Reward, which are designed for standard single-task RL, we report their fine-tuned performance on each task independently. For the lifelong RL baseline LEGION, we report the best performance achieved across its asynchronous learning process. We additionally evaluate **``Forgetting''** as defined in [3], where the scalar ranges from $0$ to $1$ ($0 = \text{no forgetting}, 1 = \text{complete forgetting}$). For our framework **LYRA**, we report performance under two settings: **$\leq 3$** number of corrections (NoC) and **$\leq 5$** NoC.
>
> The results are summarized in the table below. Our framework achieves an average success rate of 0.8 within only 3 NoC, and stably reaches 1.0 success on all tasks within 5 NoC, demonstrating **substantial efficiency compared to learning-based methods**. Notably, for simpler skills such as reach, door-open, and button-press, LYRA attains 1.0 success rate even without human intervention. Moreover, **LYRA exhibits state-of-the-art knowledge preservation** due to its external memory mechanism for structured skill storage and meta-prompting during lifelong skill functionality expansion, achieving a Forgetting score of 0.00, comparable to LEGION (0.02). Finally, although the RL baselines show reasonable performance on single-step primitive tasks, they remain fundamentally limited in handling extreme long-horizon tasks. None of them succeed on tasks such as build a house, further highlighting the necessity of non-RL, correction-efficient approaches like LYRA.
>
> **Table: Comparison with hierarchical and lifelong RL baselines on Meta-World tasks.**
> | Method               | Reach | Push | Sweep-into | Door-open | Faucet-open | Drawer-close | Button-press | Coffee-pull | Window-close | Hammer | Total average | Forgetting |
> |----------------------|-------|------|------------|-----------|-------------|--------------|--------------|-------------|---------------|--------|----------------|------------|
> | HiLL                 | 1.0   | 0.6  | 0.4        | 0.8       | 0.8         | 0.8          | 1.0          | 0.2         | 0.8           | 0.0    | 0.64           | N/A        |
> | Text2Reward          | 1.0   | 0.6  | 0.6        | 1.0       | 1.0         | 0.8          | 1.0          | 0.6         | 0.8           | 0.4    | 0.78           | N/A        |
> | LEGION               | 0.8   | 1.0  | 0.8        | 0.6       | 1.0         | 0.6          | 0.8          | 0.4         | 1.0           | 0.2    | 0.72           | 0.02       |
> | **LYRA (≤3 NoC)**    | 1.0   | 0.0  | 1.0        | 1.0       | 1.0         | 1.0          | 1.0          | 1.0         | 1.0           | 0.0    | **0.80**        | **0.00**   |
> | **LYRA (≤5 NoC)**    | 1.0   | 1.0  | 1.0        | 1.0       | 1.0         | 1.0          | 1.0          | 1.0         | 1.0           | 1.0    | **1.00**        | **0.00**   |
>
>
> In response, we add the above comparison in our revised manuscript **Appendix G, with Table S4**.
>
> Reference:
>
> [1] Zhang Q, et al. Balancing exploration and exploitation in hierarchical reinforcement learning via latent landmark graphs, International Joint Conference on Neural Networks (IJCNN). 2023.
>
> [2] Xie T, et al. Text2Reward: Reward Shaping with Language Models for Reinforcement Learning, ICLR. 2024.
>
> [3] Meng, Y., et al. Preserving and combining knowledge in robotic lifelong reinforcement learning. Nat Mach Intell 7, 256–269 (2025).

---

> > ### Author Response · Authors · 2025-11-19
> > **Response to Reviewer ak7i part 2/3**
> >
> > ### **2. Human effort in hundreds of skill code generations:**
> > We appreciate the reviewer’s observation and concern regarding the human involvement and cost required by our framework. We would like to clarify that this design is **deliberate** and central to one of our core contributions: demonstrating that **human-in-the-loop** guidance, when represented through **modular and reusable skills**, can be precisely encoded, permanently preserved, and efficiently reused to enable robots to complete extremely long-horizon manipulation tasks—a concept that has not been thoroughly explored in prior work. **To rigorously validate our hypothesis, we intentionally minimized automated LLM feedback in this work** so that our conclusions are not confounded by unstable model-generated corrections. We acknowledge that this design may increase human effort; however, having established that human feedback can be effectively distilled into a robust skill library for long-horizon control, future work can naturally incorporate **LLM–human collaborative feedback** to reduce human burden and improve overall efficiency.
> >
> > Furthermore, regarding the reviewer’s assumption that a robot must eventually master hundreds of skills, we would like to clarify that this expectation is not aligned with the current state of embodied AI. At present, whether relying on learning-based approaches or non-learning-based methods, **no AI system, including the most advanced VLA models, can reliably acquire and retain such a large number of distinct manipulation skills.** More importantly, we believe that this assumption is **unnecessary**. Due to the scarcity of large-scale, high-quality robotic data, human capabilities still far exceed those of embodied AI systems. **Even humans do not explicitly master hundreds of skills, yet this does not prevent them from functioning effectively in daily life.** Therefore, expecting robots to maintain and master hundreds of skills is neither practical nor required for real-world deployment. We believe that embodied AI systems should be designed to remain stable and effective within the specific task domains they are intended to support, thereby reducing human burden, rather than pursuing an all-encompassing capability that attempts to replace humans.
> >
> > In response, we have revised the discussion section (Page 10) to clarify that future work can naturally extend this framework with LLM–human collaborative feedback to substantially reduce human burden while retaining the advantages of modular skill representations guided by human feedback.
> >
> >
> > ---
> >
> > ## **Response to Questions:**
> >
> > ---
> >
> > ### **1. Lifelong re-evaluation and average NoC definition:**
> > **Re-evaluation:**  We appreciate the reviewer for accurately summarizing the lifelong learning process in our code generation framework. To clarify, the “learning” process in our framework refers to the agent’s ability to dynamically update the internal content of each skill function based on human feedback and environmental context. In a lifelong learning setting, this involves two requirements:  (1) preserving previously acquired skill functionality, and (2) extending skill functionality to handle new task variations. For point 1, after skill functionality is extended throughout the lifelong curriculum, **we revisit previously encountered tasks to verify whether the agent still achieves stable and consistent performance using the extended skills.** If the agent can reliably complete earlier tasks without degradation, we consider the skill to be stably preserved; this is evaluated through success rates and user confirmation *on the fly*. The evaluation for point 2 is straightforward; if the agent, after updating its skill code, can successfully solve newly introduced task variants, then the skill functionality has been successfully extended.
> >
> > **Average number of corrections:** Regarding the Number of Corrections (NoC), this metric counts **the number of human feedback steps required until the agent’s updated skill code can successfully complete a task.** It reflects the efficiency of human–agent interaction, where a lower NoC indicates more efficient learning. As reported in Figure 4 of the main text, all tasks in Meta-World and Franka Kitchen can be solved with our framework, making the success rate alone less informative. Since these benchmarks consist of multiple subtasks (each requiring only a single primitive), we report the NoC for each individual subtask. The full per-task results are provided in **Appendix D (Table S3)**, while the **average NoC** across all subtasks is summarized in the **main text, Figure 4a**. These results collectively demonstrate that our meta-prompting and modular code extension mechanisms enable efficient, stable, and extensible lifelong skill code learning and adaptation with minimal corrective intervention.

---

> > > ### Author Response · Authors · 2025-11-19
> > > **Response to Reviewer ak7i part 3/3**
> > >
> > > ### **2. The role of skills and corresponding few-shot examples:**
> > > We appreciate the reviewer for the insightful question regarding the role of *few-shot* examples in our framework. Unlike prior work that relies on in-context few-shot examples to directly generate code plans, our framework centers on the idea that temporal-invariant behaviors can be encapsulated as reusable, task-agnostic **“skills.”** As described in Figure 2 (Phase II), once a skill is well learned during lifelong learning, we store both the skill and its associated examples in the external database. **The skill itself is designed to be task (environment)-agnostic** for generalization, **while the examples are necessarily process-oriented and thus task (environment)-specific**, as they demonstrate how the same skill is applied across diverse task scenarios. These examples are therefore not used merely for adaptation through standard few-shot prompting; instead, they serve to help the agent **understand how the skill should be invoked and parameterized** in different contexts.
> > >
> > > Regarding the reviewer’s concern about **unseen tasks without corresponding examples**, our system first performs a retrieval step to determine whether any existing skill in the library can solve the new task. If no suitable skill is found (implying that also no relevant examples exist), the agent proactively proposes the creation of a new skill function head. The user then decides whether to approve and guide the **learning of this new skill**, following the *workflow* illustrated in Figure 2 (Phase I). This entire process is supported by our interactive terminal UI, which enables efficient human–agent collaboration. We refer the reviewer to our anonymous repository for implementation details.
> > >
> > > In response to the reviewer’s questions, we have revised the method section to clarify exactly how in-context learning operates within our framework and how examples are used in conjunction with skills (Page 5). Additionally, we have expanded Appendix B with UI snapshots (Fig. S1) and detailed explanations of how the interactive interface supports dynamic example retrieval, new skill proposal, and user-guided refinement throughout lifelong learning.
> > >
> > > ---
> > >
> > > ### **3. Skill function and environment attribute setup:**
> > > We appreciate the reviewer’s question regarding the level of detail required in user instructions during skill creation and attribute specification. As illustrated in Figure 2 (Phase I), when learning a new skill, the agent first proposes a candidate function head, including initial parameter attributes, based on the user’s high-level **free-form text instruction**. For example, to learn a block-stacking skill, a user might simply say, **“learn a skill that stacks the blocks.”** The user then reviews the function head suggested by the LLM agent and may either accept it or provide corrective feedback (e.g., requesting an additional orientation input). The final structure of the skill function head is determined through this **multi-round, free-form natural-language dialogue**, without requiring users to manually specify implementation details.
> > >
> > > Importantly, users are **not required to have precise or detailed knowledge of environment attributes**. The only expectation is basic common-sense knowledge. For instance, knowing that a rigid object’s pose can be represented by a 6-DoF vector. With real-time rendered frames provided by the simulator, users can typically distinguish fundamental object characteristics such as shape (cube vs. cylinder), size, color, and coarse spatial relationships (left/right, near/far). This level of qualitative understanding is sufficient.
> > >
> > > In response to the reviewer’s comment, we have revised the method section in the manuscript (Page 5) to further clarify the nature of user instructions, the interactive refinement process for creating functions, and the minimal knowledge required from the user.

---

> ### Author Response · Authors · 2025-11-30
> **Summary of Reviewer ak7i’s comments and our responses**
>
> Below, we summarize the reviewer’s main concerns and our corresponding responses:
>
> ---
>
> ## **Reviewer’s Main Points**
> Reviewer ak7i raised the following key issues:
>
> - Lack of comparisons with hierarchical or lifelong RL baselines.
> - Human effort cost due to user-guided skill code generation, curriculum design, and hinting.
> - Need for clarification on:
>    - Lifelong re-evaluation, meta-prompting, and the definition of NoC (Number of Corrections).
>    - The role and robustness of few-shot examples.
>    - Whether examples are environment-agnostic or environment-specific.
>    - How skill functions and environment attributes are specified.
>    - How detailed user instructions must be.
> ---
>
> ## **Our Responses**
>
> ### **1. Added lifelong and hierarchical RL baselines comparison.**
>   We conducted additional evaluations on 10 Meta-World tasks using HiLL, Text2Reward, and LEGION.
>   - ALL existing SOTA RL baselines fail on long-horizon tasks such as *build-a-house*.
>   - For Meta-World tasks, our method achieves up to 1.0 success with stable performance under ≤5 NoC.
>   - Results are provided in the revised manuscript Appendix G, Table S4.
>
> ### **2. Clarification of human effort.**
>   **Our design intentionally minimizes automated feedback** so that the central hypothesis can be evaluated without unstable model-generated corrections.
>   We further clarify that no current embodied AI system can reliably master hundreds of distinct skills, and that future extensions with LLM–human collaborative feedback can reduce human effort.
>
> ### **3. Lifelong re-evaluation and NoC definition clarified.**
>   - Re-evaluation ensures that previously acquired skills remain stable when new skills are added.
>   - NoC quantifies the number of human corrections required before an updated skill reliably solves the task.
>   - Results are provided in revised Appendix D (Table S3) and Figure 4a.
>   - Role and design of few-shot examples explained.
>     - Skills are task-agnostic; examples are task-specific demonstrations of how skills are invoked and parameterised.
>     - Examples are not used as standard few-shot prompts, but to show the agent how a skill should be instantiated in different contexts.
>     - For unseen tasks with no matching examples, the agent proposes a new skill through the interactive UI.
>   - Skill function creation and environment attributes clarified.
>     - Users provide high-level free-form text, and the agent proposes a candidate function head.
>     - Users refine the function through natural-language dialogue; only basic common-sense knowledge is required.
>     - Full UI workflow and examples are included in revised Appendix B (Fig. S1).
>
> ---
>
> ## **Conclusion**
> We have addressed ALL concerns raised by Reviewer ak7i through new experiments, expanded analysis, clarified methodology, and detailed appendix documentation.
> These updates resolve the reviewer’s questions and reinforce the contributions and reliability of our lifelong skill code generation framework.

---

> ### Author Response · Authors · 2025-11-30
>
> We have addressed ALL of the reviewer’s concerns with point-by-point responses and believe that our clarifications and revisions fully resolve the issues raised. We look forward to further feedback from the reviewer and the area chair.

---

### Official Review · Reviewer_njQ7 · 2025-11-01

**Soundness:** 3
**Presentation:** 4
**Contribution:** 3
**Rating:** 8
**Confidence:** 3

**Summary:**

This paper aims to address limitations in using LLMs for robotic code generation, specifically their struggles with long-horizon tasks, language ambiguity, and catastrophic forgetting. The authors propose LYRA, a human-in-the-loop (HITL) lifelong learning framework. LYRA enables an agent to learn reusable skills as Python functions, guided by human feedback. These skills and successful task examples are stored in an external memory and retrieved using Retrieval-Augmented Generation to solve new tasks. The framework includes a user-designed curriculum for extending skill capabilities and a "hint" mechanism for guiding retrieval. Experiments in simulation (Ravens, Franka Kitchen, MetaWorld) and on a real Franka FR3 robot show LYRA achieves relatively strong performance and improvement in feedback efficiency, successfully solving complex tasks like "build a house".

**Strengths:**

- Involving HITL feedback, a user-driven curriculum to LLM-robotic control is novel and interesting.
- The evaluation is thorough, spanning three distinct simulation benchmarks (Ravens, Franka Kitchen, MetaWorld) and a real-world Franka FR3 robot.
- The ablations are insightful: LYRA w/o memory and LYRA w/ LLM feedback directly prove the value of the two core components.
- The "build a house" task serves as a good and compelling demonstration. Claiming to be the first to solve this task, the authors clearly show how a complex behavior can be decomposed into 12 hierarchically-learned skills.

**Weaknesses:**

- The paper combines several well-established ideas: HITL, RAG, and code-generation. The authors should clarify whether the framework introduces specific novel contributions versus the integration of existing ideas.

- The paper strongly motivates the need for HITL by showing LLM-only feedback is unreliable, but it does not adequately quantify the cost of the human. The "42% efficiency improvement" is measured in "Average number of corrections". This metric is useful but incomplete. It hides the human's cognitive load and time-per-round. A human correction might be 1 round but take 10 minutes, while an LLM correction takes 10 rounds at 30 seconds each.

**Questions:**

- The "hint" mechanism is described as "simple yet effective", but its implementation and UI are omitted. How does a user provide this hint? Is it free-form text that is then parsed?

- What is the mechanism for the "roll back from failure"? If a 20-step plan (like "build a house")  fails at step 15, does the user correct from that state, or must the entire task be restarted? How is task state managed?

---

> ### Author Response · Authors · 2025-11-19
> **Response to Reviewer njQ7 part 1/2**
>
> ## **Response to Weakness:**
> ---
> ### **1. Clarification of novel contribution**
> We appreciate the reviewer’s comment and the opportunity to clarify the novelty of our framework. While we acknowledge that our system incorporates components such as HITL, RAG, and code generation, **the core novelty of this work is conceptual rather than purely architectural or engineering-based**. Specifically, we demonstrate that **human feedback can be systematically distilled, stored, and efficiently reused in the form of modular “skills”**, enabling a robot to accumulate persistent knowledge that supports **extremely long-horizon manipulation tasks** far beyond the reach of current methods. This idea, in which human effort can be transformed into stable, generalizable, and compositional skill representations within an LLM-driven code-generation pipeline, has not been thoroughly explored or validated in prior work. **The tools we employ serve only as mechanisms for acquiring these robust skills and ensuring that they remain reliable throughout lifelong use.** Moreover, as discussed in the paper, fully automated closed-loop LLM/VLM systems struggle with long-horizon reasoning due to error accumulation and limited context depth; thus our **deliberate design, which minimizes automated feedback, allows us to rigorously isolate and validate the central hypothesis.** Having established that human-derived skill representations can indeed support long-horizon tasks, future work can extend this conceptual contribution by integrating LLM–human collaborative feedback to reduce intervention cost while retaining the benefits of our skill-based formulation.
>
> In response to the reviewer’s comment, we have further revised and refined the introduction section (Pages 2 and 3) to clearly highlight and emphasize the core novelty of our paper.
>
> ---
> ### **2. Quantification of human cost**
> We appreciate the reviewer for highlighting the importance of quantifying human time cost and cognitive load. To address this concern, we analyzed timestamped logs from **30 feedback loops collected in the Ravens benchmark, data collected from the same user**. Our interactive terminal UI allows users to input free-form text in real time, enabling precise measurement of the full human feedback cycle. Because the interface provides predefined interaction options, we use three timestamps as reference points:
> (1) $t_1$, the moment the task execution finishes;
> (2) $t_2$, the moment the user selects the “Provide Feedback” option; and
> (3) $t_3$, the moment the feedback is submitted.
> The interval $t_2 - t_1$ represents the user’s reasoning and decision-making time, and $t_3 - t_2$ represents the time required to type and submit the feedback. The results are listed in the table below, which shows **no order-of-magnitude difference between human and LLM feedback time**. Human feedback, including both thinking and typing, averages around 30 seconds, while LLM feedback averages 15.8 seconds. Importantly, the ranges of human and LLM feedback times overlap, indicating that the differences are incremental rather than fundamental.
>
> **Table: Comparison of human-in-the-loop and LLM feedback time efficiency.**
> |                          | **Human user**                                   | **LLMs**                                  |
> |--------------------------|--------------------------------------------------|-------------------------------------------|
> | Average time of a feedback loop | 30.5 sec (Min 17.2 sec, Max 39.5 sec) | 15.8 sec (Min 4 sec, Max 32.4 sec)        |
> | Average time for reasoning/decision | 8 sec (below 25%)                | 14 sec (over 90%)                         |
> | Average time for text typing/generation | 22 sec (over 75%)             | 2 sec (below 10%)                         |
>
> When examining decision-making time independently, we find that humans are often faster than LLMs. By segmenting timestamps at the moment the “provide feedback” option is selected, we observe that humans typically require **less than 8 seconds** to determine the correct feedback. **The majority of human time cost comes from typing, which varies with user typing proficiency and does not reflect cognitive burden**. In contrast, LLMs spend little time on token generation (under 10% of total time) but still require time for model inference and reasoning. These findings suggest that **human cognitive load is relatively low, and the time difference is largely attributable to mechanical typing rather than conceptual difficulty**.

---

> ### Author Response · Authors · 2025-11-19
> **Response to Reviewer njQ7 part 2/2**
>
> Although LLMs are somewhat faster in feedback generation, they are substantially more limited in the structure and expressiveness of the feedback they can produce. Prior methods such as DAHLIA [1] require feedback in a rigid, predefined format (binary success indicators, fine-grained object/pose/action tuples), which restricts their generalization to complex or novel scenarios. In contrast, our framework supports **free-form human feedback to provide more informative, more flexible guidance**. While this introduces a modest additional time cost, our experiments demonstrate that **such flexibility is essential for enabling the agent to escape local optima and learn robust, reusable skills**, particularly in extreme long-horizon tasks where LLM-only feedback consistently fails.
>
> Reference:
>
> [1] Meng Y, Yao X, Ye H, et al. Data-agnostic robotic long-horizon manipulation with vision-language-guided closed-loop feedback[J]. arXiv preprint arXiv:2503.21969, 2025.
>
> ---
> ## **Response to Questions:**
>
> ---
>
> ### **1. Explanation of ``hint'' mechanism**
>
> We appreciate the reviewer for raising this question and agree that additional clarification on the “hint” mechanism is valuable. In our framework, the hint is indeed provided as **free-form text** through the **interactive terminal UI** (snapshots see our revised Appendix B, Fig. S1). During code generation, the agent retrieves potentially relevant skills and examples from the external database by computing cosine similarity between the embeddings of skill names and the task context. However, because our skill library is developed bottom-up while long-horizon tasks require top-down decomposition, **this automatic retrieval may occasionally miss a relevant skill**. The hint mechanism offers a simple yet highly effective solution: the user can type a **natural-language hint** such as **“You might need to use the 'move_next_to_reference' skill to build the house base.”**  This lightweight intervention informs the agent that a particular existing skill should be considered, significantly improving retrieval accuracy in HITL settings. Although more advanced RAG retrieval methods could further optimize this process in future work, the current mechanism already performs well in practice.
>
> To make this clearer, we have revised the manuscript to describe the implementation and interaction flow of the hint mechanism (Page 5), and we have added a snapshot and detailed explanation of the terminal UI in the updated Appendix B, Fig. S1.
>
> ---
>
> ### **2. Explanation of ``roll back from failure'' and task state management**
>
>  **Regarding the mechanism for “rollback from failure”**. Users can observe the entire execution process in real time through the simulation stream, allowing them to quickly determine the point and cause of failure. Our framework handles different failure modes accordingly:
>
> 1. **If failure results from stochastic noise** in randomized environment initialization (e.g., improper object initial position), we record it as a statistical failure and simply rerun the task with the same task code plan.
>
> 2. **If failure exposes a deficiency in an existing skill**, we halt execution and return to Phase II of Figure 2 to refine or extend the skill through lifelong learning until the skill becomes robust.
>
> 3. **If failure occurs because the agent generates behavior outside the distribution of all existing skills**, which indicates that a new skill is required. The user may consider terminating the current run and returning to Phase I of Figure 2 to initiate learning of that new skill.
>
> 4. **If task execution fails in Phase I or Phase II** because the agent-produced skill is insufficient, the skill is updated using human feedback, and the task is subsequently reattempted.
>
> As emphasized in the main text, **the workflow in Figure 2 is not linear**; users may flexibly transition between phases, starting or stopping any stage as needed during long-horizon task execution.
>
> **Regarding task state management**. Our customized Ravens settings inherit environment construction and **API** mechanisms from the original Ravens and LoHoRavens benchmarks. Before execution, an LLM-based environment manager configures the scenario based on the user’s free-form instructions (e.g., “Generate a scenario with four differently colored blocks”), selects appropriate **STL models** from the asset library, and loads them into the workspace. During task execution, the code-generation agent obtains necessary scene information through predefined environment APIs, **such as object names, positions, and bounding boxes**. For Meta-World and Franka Kitchen, we skip scene generation and directly provide control APIs such as `movep` and `gripper_action`, along with environment-query APIs like `get_obj_name` and `get_obj_pos`, consistent with Ravens-style interfaces. All APIs used for state retrieval and interaction are documented in **Appendix B** and included in the anonymous repository.

---

> ### Author Response · Authors · 2025-11-30
> **Summary of Reviewer njQ7’s comments and our Responses**
>
> Below, we summarize the reviewer’s main concerns and our corresponding responses:
>
> ---
>
> ## **Reviewer’s Main Points**
> - **Novelty clarification:** The reviewer requested clearer justification of what is novel beyond combining existing components like HITL or RAG.
> - **Human cost quantification:** The reviewer noted that our claim of “42% improvement” does not quantify the full human cognitive load and asked for a concrete timing analysis.
> - **“Hint” mechanism explanation:** The reviewer found the mechanism useful but inadequately described, and asked how users actually provide hints.
> - **Rollback mechanism:** The reviewer asked how “rollback from failure” operates and how task state is managed in long-horizon execution.
>
> ---
> ## **Our Responses**
>
> ### **1. Novelty clarification**
> - We clarified that **our contribution is primarily conceptual**, not engineering-based or incremental.
> - We show that **human feedback can be distilled, stored, and reused as modular skills**, enabling stable compositional behaviors in long-horizon tasks.
> - HITL, RAG, and code generation are **mechanisms**, not the novelty itself.
> - We revised the introduction (Pages 2–3) to highlight this conceptual contribution.
>
> ### **2. Quantification of human cost**
> - We analyzed **30 feedback loops** collected in the Ravens benchmark, all from the same user.
> - We separated the feedback loop into precise timestamps \(t_1, t_2, t_3\) to measure:
>   - Reasoning time \(t_2 - t_1\)
>   - Typing time \(t_3 - t_2\)
> - Results show:
>   - Human feedback loop: **30.5 seconds** on average
>   - LLM-only feedback: **15.8 seconds** on average
>   - Human reasoning time: **less than 8 seconds**, often faster than LLMs
> - Conclusion: Human cognitive load is **low**, with most time cost attributable to mechanical typing.
> - These findings address the reviewer’s concern that human cost was previously under-measured.
>
> ### **3. Explanation of the “hint” mechanism**
> - We clarified that hints are **free-form text** entered through our interactive terminal UI (with UI snapshots added to Appendix B, Fig. S1).
> - Because bottom-up skill libraries may miss a relevant skill, the user can provide a natural-language hint to guide retrieval.
> - This lightweight mechanism significantly improves retrieval reliability in practice.
>
> ### **4. Rollback mechanism and task state management**
> - We detailed the four types of failure handling:
>   1. Random environment noise
>   2. Deficiency in existing skill
>   3. Behavior outside the distribution of existing skills
>   4. Failure during ongoing learning phases
> - We clarified that the workflow in Figure 2 is **not linear** and that users may transition between phases as needed.
> - We described how task state is managed using Ravens-style environment APIs, with all interaction APIs listed in Appendix B.
>
> ---
>
> ## **Conclusion**
> - Reviewer njQ7’s concerns led to **clearer clarifications and new additional quantitative analyses**, improving both transparency and rigor.
> - We have now:
>   - Clearly articulated the **core conceptual novelty** of our framework.
>   - Provided **concrete timing analysis** demonstrating low human cognitive load.
>   - Added detailed descriptions of the **hint mechanism**, **rollback logic**, and **task state APIs**.
> - These additions directly address all points raised by the reviewer and strengthen the clarity and completeness of the submission.

---

> ### Author Response · Authors · 2025-11-30
>
> We have addressed ALL of the reviewer’s concerns with point-by-point responses and believe that our clarifications and revisions fully resolve the issues raised. We look forward to further feedback from the reviewer and the area chair.

---

### Author Response · Authors · 2025-11-19
**Summary of Responses to the Reviewers, Area Chairs, and Program Chairs**

We would like to express our sincere gratitude to the reviewers for their thoughtful and constructive feedback throughout the review process, which has been invaluable in improving the clarity, rigor, and overall quality of our submission. We have carefully addressed every comment raised by the reviewers and have incorporated all requested empirical and quantitative analyses, including

**(1) Clarification of novelty contribution. (Response to Reviewers njQ7 and rJdU)**
The **core novelty of our work is conceptual rather than engineering-based or incremental**: we show that human feedback can be systematically distilled into modular, reusable **skill** representations that persist throughout lifelong use and enable reliable control of extremely long-horizon manipulation tasks. This insight, which transforms human effort into stable and compositional skill structures within an LLM-driven code-generation pipeline, to the best of our knowledge, **has not been thoroughly explored or validated in prior work**. Our intentional minimization of automated feedback allows us to rigorously isolate and validate this hypothesis, with components such as RAG and HITL serving only as tools for acquiring and maintaining robust skills and ensuring that they remain reliable throughout lifelong use.
For these reasons, we respectfully note that the **characterization by Reviewer rJdU that our contribution is merely a simple combination of existing techniques is incomplete and does not accurately reflect the conceptual advancement presented in this work.** We therefore **disagree** with this claim and hope that our clarifications help convey the originality of our contribution.

**(2) The quantitative assessment of HITL time efficiency. (Response to Reviewers njQ7 and rJdU)**
Our timestamp analysis shows that human reasoning time remains under 8 seconds and the full feedback cycle averages ~30 seconds, which is of the same order of magnitude as LLM-based feedback (15.8 seconds on average); this demonstrates that **human-in-the-loop intervention introduces no substantial time penalty while providing far more reliable guidance.**

**(3) Comparisons with recent hierarchical or lifelong RL baselines. (Response to Reviewer ak7i)**
Across 10 Meta-World tasks, **LYRA** achieves up to **1.0 success rate** with $\leq 5$ corrections and attains zero forgetting, **outperforming** HiLL, Text2Reward, and LEGION, all of which fail to generalize to extreme **long-horizon tasks**. It is also worth highlighting that, unlike traditional learning-based approaches, our code-generation method **requires no neural network training**. This allows the system to acquire stable, reusable, and strongly generalizable skills **within minutes**. In contrast, learning-based methods such as RL typically require more than a million training steps and many hours, sometimes tens of hours, to produce reasonable behaviors, yet they often still struggle with long-horizon tasks. Although this comparison is not central to our main argument, it further highlights the efficiency and reliability of our framework.

**(4) Comparisons with recent VLA models. (Response to Reviewer rJdU)**
Across various benchmarks, while OpenVLA and $\pi_0$ perform comparably on short-horizon tasks (e.g., Metaworld tasks), both collapse to **0.0 success** on our **$>$20-primitive Ravens tasks**, whereas our **LYRA** maintains **0.80–0.95 success**, demonstrating **clear outperformance in extreme long-horizon manipulation.**
It is important to note that the comparison requested by the Reviewer rJdU between learning-based methods and non-learning-based methods does not provide meaningful insight in practice. Our code-generation approach follows a **fundamentally different technical route from learning-based methods**, and therefore should not be interpreted as a direct competitor. Instead, **our method can serves as a complementary approach to imitation-learning-based VLA models**. By using our framework, one can rapidly obtain large-scale, high-quality, and generalizable long-horizon trajectories in a reliable manner. These trajectories can, in turn, support and enhance the training and performance of VLA models on long-horizon tasks.

Detailed responses to each point are provided in our individual rebuttals. **For each reviewer, we have also added a concise summary of both the review points and our corresponding responses at the end of the detailed comments, which helps the Area Chair quickly review the key information.** We are deeply grateful to the reviewers, the Area Chairs, and the Program Chairs for their time, effort, and insightful guidance. We respectfully look forward to the reviewers' further feedback on our revised manuscript and response comments, and we remain committed to contributing meaningfully and responsibly to the open-source AI and robotics community.

---

### Author Response · Authors · 2025-11-26
**Kind Reminder: Follow-Up Before Discussion Period Ends**

Dear Reviewers,

We hope this message finds you well. As the discussion period is nearing its end with less than one week remaining, we want to ensure we have addressed all your concerns satisfactorily. If there are any additional points or feedback you'd like us to consider, please let us know. Your insights are invaluable to us, and we're eager to address any remaining issues to improve our work.

Thank you for your time and effort in reviewing our paper.

Authors of Paper No.1431

---

### Meta-Review · Area_Chair_48m1 · 2026-01-06

**Summary:**

This paper proposes LYRA, a human-in-the-loop lifelong code-generation framework where human feedback is distilled into modular, reusable skill functions stored in external memory and retrieved via RAG and lightweight hints to solve long-horizon robotic manipulation tasks that exceed the reach of standard LLM-only pipelines.

Reviewers appreciated the move towards persistent, interpretable skill representations, and the integration of human-in-the-loop (HITL) feedback, external memory, and modular skills (rather than pure end-to-end LLM control). They also appreciated the empirical performance on long-horizon tasks like "build a house".

They also pointed out the following key issues:
- The technical novelty is limited: the system primarily integrates existing ideas (HITL, RAG, code).
- Concerns about scalability of the HITL feedback and human-designed curriculum, level of expertise required etc.
- Missing or weak comparisons to strong baselines (hierarchical/lifelong RL, VLAs).
- Insufficient detail on hint mechanism and retrieval.
- Insufficient detail on rollback from failure and task state management.

**Reviewer Concerns:**

Addressed satisfactorily:
- Missing or weak comparisons to strong baselines (hierarchical/lifelong RL, VLAs). Rebuttal adds Meta-World comparisons to HiLL, Text2Reward, LEGION showing LYRA is superior; adds VLA comparisons showing OpenVLA/π0 collapse on long-horizon tasks; clarifies that those methods are complementary rather than direct competitors.

- Insufficient detail on hint mechanism and retrieval. Rebuttal provides missing details: hints are free-form text.

- Insufficient detail on rollback from failure and task state management. Rebuttal provides missing details.

Partially addressed / unaddressed:
- Concerns about scalability of the HITL feedback and human-designed curriculum, level of expertise required etc. Rebuttal provides timestamp analysis showing ~30.5s per human feedback loop with <8s reasoning; argues cognitive load is low, typing dominates; frames HITL as deliberate core of contribution and suggests future LLM–human collaboration to reduce burden; clarifies realistic number of skills and total hours (~3.3h for Meta-World library). Claims no expertise required. However, for a paper that claims human-in-the-loop as central to its contributions, the rigor of the human subjects study leaves a lot of room for improvement. it does not specify who the users were, how many participated, what level of expertise they possessed, or whether results reflect a single operator or multiple individuals. The absence of this information makes it difficult to assess reproducibility, the true human burden, and the robustness of the approach to variation in user skill or interaction style. Given that the framework’s core contribution relies on human feedback to construct and refine the skill library, clearer reporting of user demographics, expertise, and inter‑user variability would strengthen the empirical claims and help readers evaluate the scalability and generality of the proposed method.

- The technical novelty is limited: the system primarily integrates existing ideas (HITL, RAG, code).  Authors agree, but point to the conceptual novelty of distilling human feedback into skills. This is a subjective point, and I'm largely on board with the authors' point; reviewers might not have been convinced.

**Reviewer Scores:**

My assessment overall is that the discussion would have highlighted the limited nature of the human subjects-centric evaluation study, on which nearly all the core claims hinge. This is also the dominant negative factor in my evaluation.

njQ7: 8->6
ak7i: 4->4
rJdU: 4->4

---

### Decision · Program_Chairs · 2026-01-26

Reject